# The Limitations of Large Width in Neural Networks: A Deep Gaussian Process Perspective

**Geoff Pleiss**
Columbia University
gmp2162@columbia.edu

**John P. Cunningham**
Columbia University
jpc2181@columbia.edu

## Abstract

Large width limits have been a recent focus of deep learning research: modulo computational practicalities, do wider networks outperform narrower ones? Answering this question has been challenging, as conventional networks gain representational power with width, potentially masking any negative effects. Our analysis in this paper decouples capacity and width via the generalization of neural networks to Deep Gaussian Processes (Deep GP), a class of nonparametric hierarchical models that subsume neural nets. In doing so, we aim to understand how width affects (standard) neural networks once they have sufficient capacity for a given modeling task. Our theoretical and empirical results on Deep GP suggest that *large width can be detrimental to hierarchical models*. Surprisingly, we prove that even nonparametric Deep GP converge to Gaussian processes, effectively becoming shallower without any increase in representational power. The posterior, which corresponds to a mixture of data-adaptable basis functions, becomes less data-dependent with width. Our tail analysis demonstrates that width and depth have opposite effects: depth accentuates a model's non-Gaussianity, while width makes models increasingly Gaussian. We find there is a "sweet spot" that maximizes test performance before the limiting GP behavior prevents adaptability, occurring at width = 1 or width = 2 for nonparametric Deep GP. These results make strong predictions about the same phenomenon in conventional neural networks trained with L2 regularization (analogous to a Gaussian prior on parameters): we show that such neural networks may need up to $500 - 1000$ hidden units for sufficient capacity—depending on the dataset—but further width degrades performance.

## 1 Introduction

Research has shown that deeper neural networks tend to be more expressive and efficient than wider networks under a variety of metrics [e.g. 21, 63, 67, 70, 74, 75, 78, 83]. Nevertheless, there is resurgent interest in wide models due in part to empirical successes [e.g. 92] and theoretical analyses of limiting behavior. When randomly initialized to create a distribution over functions, neural networks converge to Gaussian processes (**GP**) as width increases. This result, first proved for 2-layer networks [69], has been extended to deeper networks [56, 64], convolutional networks [38, 71], and other architectures [50, 88]. A similar limit exists for gradient-trained networks, which behave increasingly like kernel machines under the neural tangent kernel [e.g. 6, 8, 28, 39, 52, 57, 89].

While these limits simplify analyses, there is something unsettling about reducing neural networks to kernel methods. Neal [69, p. 161] describes the GP limit as "disappointing," noting that "infinite networks do not have hidden units that represent 'hidden features'... often seen [as the] interesting aspect of neural network learning." Recent work indeed shows that learned hierarchical features can be exponentially more efficient than the fixed shallow representations of kernels [e.g. 4, 5, 8, 11, 13, 21, 41, 42, 60, 91]. At the same time, wider networks can more accurately model complex functions

35th Conference on Neural Information Processing Systems (NeurIPS 2021).

[44]. Thus, wide limits appear to confound opposing phenomenon: increased capacity makes them more expressive, yet the loss of hierarchical features seems to make them less expressive. This may explain the mixed empirical performance of limiting models: outperforming finite width models in some scenarios [e.g. 9, 38, 39, 58], yet falling short on more complex tasks [e.g. 8, 11, 35, 57, 81].

This paper aims to decouple these effects of large width. Our goal is to understand the inductive biases of wide networks, after a network has "sufficient" capacity for a given modeling task. We ask: *If we control for the effects of increased capacity, what—if any—value remains in wide networks?*

To achieve this control, we note that a typical neural network layer corresponds to a finite basis, where elementwise nonlinearities transform each hidden feature into a *single basis function*. In order to decouple width from capacity, one could generalize these layers so that each nonlinearity produces any number of basis functions; if each hidden feature gives rise to an infinite and universal basis, then hidden layers would have infinite representational capacity *regardless of width*. This generalization is in fact a well-studied class of hierarchical models—Deep Gaussian Processes (**Deep GP**) [19, 24, 26, 27, 30, 32, 46, 79]—where standard neural net layers are replaced with vector-valued Gaussian processes. Indeed, typical neural networks are a degenerate Deep GP subclass [1, 2, 33, 72].

We therefore have a generalization of neural networks where capacity is controlled, from which we can glean insights about conventional networks that have sufficient representational power for a given modeling task. Surprisingly, despite using Gaussian processes as the primary hierarchical component, we prove that *Deep GP converge to (single-layer) GP in their infinite width limit* (Thm. 1). Troubling implications immediately ensue: large width is strictly detrimental to Deep GP, as the limiting model collapses to a shallower version of itself. We support this theorem with an analysis of neural network and Deep GP posteriors, which *become less adaptable as width increases*. Specifically, we show that the posterior mean corresponds to a mixture of functions drawn from data-dependent (and thus adaptive) reproducing kernel Hilbert spaces, formalizing the above claim from Neal [69]. As width increases, this mixture collapses to the data-independent kernel of the limiting GP, implying that wider models have less feature learning. Finally, we present a novel tail analysis which indicates that *width and depth have opposite effects*: depth accentuates non-Gaussianity, sharpening peaks and fattening tails, whereas width increases Gaussianity (Thms. 2 and 3).

Our theoretical results hold for Deep GP and conventional (parametric) neural networks alike. Experiments confirm that—after a model achieves sufficient capacity[1]—*width can become harmful to model fit and performance*. For nonparametric Deep GP, a width of 1 or 2 often achieves the best performance. Neural networks—because of their parametric nature—naturally require more hidden units before achieving optimal accuracy. Nevertheless, for Bayesian neural networks and conventional (optimized) neural networks trained with L2 regularization, performance degrades after a certain width. On small datasets ($N \leq 1000$) with low dimensionality, we find that models with $\leq 16$ hidden units achieve best test set performance. On larger datasets like CIFAR10, this "sweet spot" occurs later (at $\approx 500$ hidden units for sufficiently deep models), yet performance degrades beyond this width. We note that these trends do not necessarily hold for models that do not have a probabilistic interpretation—i.e. optimized neural networks trained without (or nearly without) L2 regularization. Nevertheless, our findings suggest that narrower models have better inductive biases, and wide models perform well *in spite of*—not because of—large width.

## 2  Setup

### 2.1  Related Work

**Effects of width.** Works have shown that, given finite parameters, deeper models are more expressive than wider models [63, 67, 74, 75, 83]. Similarly to our work, Aitchison [2] recognises the link between finite neural networks and Deep GP, and argues that finite neural networks have flexibility in the top-layer representation that is absent in the infinite-width limit. Halverson et al. [45] draw a connection to quantum field theory to argue that neural networks become "simpler" near their infinite-width limit. In the non-probabilistic setting, it is worth noting that wide models have been shown to have favorable optimization landscapes [7, 28, 59, 70, 82] and are resistant to overfitting via double descent [12, 20, 68]. Our work controls for these factors by examining nonparametric hierarchical models with exact Bayesian inference, and thus does not disagree with these other works.

---

[1]We offer a formal notion of "sufficient capacity" in Appx. B.5.

Infinite width limits have received renewed interest in Bayesian [38, 50, 57, 64, 69, 71, 88] and non-Bayesian [6, 8, 22, 28, 43, 52, 57, 65, 89, 90] settings. Most of these works show that neural networks converge to kernel methods, though recent work suggests that this limiting behavior can be avoided with different parameterizations [e.g. 22, 43, 65, 90]. Similarly to Lee et al. [57], our Deep GP limit analysis sequentially increases the width of each layer, though we hypothesize a similar proof exists where the width of all layers increases simultaneously (akin to [64]).

**Deep GP** are introduced by Damianou and Lawrence [27]. A large portion of Deep GP research has thus far focused on scalable approximate inference methods [19, 24, 25, 32, 46, 72, 79, 85]. Though prior work has studied tail properties of neural networks [84, 93] and Deep GP with RBF kernels [62], our work is—to the best of our knowlege—the first general result for Deep GP tails. Duvenaud et al. [33] and Dunlop et al. [29] investigate pathological behaviors that arise with depth, while Agrawal et al. [1] note that "bottlenecked" Deep GP have better performance and correlations among predictive tasks. Our work complements these analysis by characterizing the effects of width.

**Connections between Deep GP and neural networks.** Many researchers have noted connections between neural networks and Deep GP [e.g. 24, 31, 36, 61]. Duvenaud et al. [33] suggest that infinitely-wide neural networks with intermediate bottleneck layers are nonparametric Deep GP. Agrawal et al. [1] formalize this connection, but note that not all Deep GP can be constructed from bottlenecked neural networks (see Appx. E). In contrast to these prior works, we avoid reducing Deep GP to neural networks, and instead reduce neural networks to degenerate Deep GP.

## 2.2 A Covariance Perspective on Gaussian Process Limiting Behavior

To decouple the effects of increasing width and capacity, we first prove a new result about GP limits for a more general class of models, including Deep GP as well as typical neural networks. This result forms a necessary foundation for the subsequent theorems that are a main contribution of this work. To begin, note that the proof technique introduced by Neal [69] and extended by others [38, 50, 56, 64, 71, 88] relies on the multivariate central limit theorem, which requires a model with additive structure. Deep GP do not generally decompose in an additive manner, so we establish a more general proof technique. For simplicity, we first present it in the context of neural networks, and then extend it to a more general class of models.

Consider the 2-layer neural network $f_2(\mathbf{f}_1(\mathbf{x}))$, with $\mathbf{f}_1 : \mathbb{R}^D \to \mathbb{R}^{H_1}$ and $f_2 : \mathbb{R}^{H_1} \to \mathbb{R}$:

$$\mathbf{f}_1(\cdot) = \mathbf{W}_1^\top(\cdot) + \beta\mathbf{b}_1, \qquad f_2(\cdot) = \tfrac{1}{\sqrt{H_1}}\mathbf{w}_2^\top\boldsymbol{\sigma}(\cdot) + \beta b_2. \tag{1}$$

$\boldsymbol{\sigma}(\cdot)$ is an elementwise nonlinearity, $\beta$ is a positive constant, and $\mathbf{W}_1$, $\mathbf{b}_1$, $\mathbf{w}_2$, and $b_2$ are i.i.d. Normal. With randomly initialized parameters, $f_2(\mathbf{f}_1(\cdot)) : \mathbb{R}^D \to \mathbb{R}$ is a prior distribution over functions, and this distribution converges to a GP in the infinite width limit [69].

**Lemma 1.** *The neural network defined in Eq. (1) is a Gaussian process if and only if—for any finite set of inputs $\mathbf{X} = [\mathbf{x}_1, \ldots, \mathbf{x}_N]$—the conditional prior covariance $\mathbb{E}_{\mathbf{f}_2|\mathbf{X},\mathbf{W}_1,\mathbf{b}_1}[\mathbf{f}_2\mathbf{f}_2^\top]$ is almost surely equal to the marginal prior covariance $\mathbb{E}_{\mathbf{f}_2|\mathbf{X}}[\mathbf{f}_2\mathbf{f}_2^\top]$, where $\mathbf{f}_2 \mid \mathbf{X} \triangleq [f_2(\mathbf{f}_1(\mathbf{x}_1)), \ldots, f_2(\mathbf{f}_1(\mathbf{x}_N))]$.*

*Proof.* By definition, $f_2(\mathbf{f}_1(\cdot))$ is a GP if and only if $\mathbf{f}_2 \mid \mathbf{X}$ is multivariate Gaussian for any $\mathbf{X}$. From Eq. (1), we have $p(\mathbf{f}_2 \mid \mathbf{X}, \mathbf{W}_1, \mathbf{b}_1) = \mathcal{N}(\mathbf{0}, \mathbf{K}_{\mathbf{W}_1,\mathbf{b}_1}(\mathbf{X},\mathbf{X}))$, where $[\mathbf{K}_{\mathbf{W}_1,\mathbf{b}_1}(\mathbf{X},\mathbf{X})]_{ij} = \beta^2 + \frac{1}{H_1}\boldsymbol{\sigma}(\mathbf{W}_1^\top\mathbf{x}_i + \beta\mathbf{b}_1)^\top\boldsymbol{\sigma}(\mathbf{W}_1^\top\mathbf{x}_j + \beta\mathbf{b}_1)$ is the appropriate kernel Gram matrix. Using Jensen's inequality, we have a lower bound on the characteristic function of $\mathbf{f}_2 \mid \mathbf{X}$:

$$\mathbb{E}_{\mathbf{f}_2|\mathbf{X}}\left[\exp\left(i\mathbf{t}^\top\mathbf{f}_2\right)\right] = \mathbb{E}_{\mathbf{W}_1,\mathbf{b}_1}\left[\mathbb{E}_{\mathbf{f}_2|\mathbf{X},\mathbf{W}_1,\mathbf{b}_1}\left[\exp\left(i\mathbf{t}^\top\mathbf{f}_2\right)\right]\right] \qquad \text{(law of total expectation)}$$

$$= \mathbb{E}_{\mathbf{W}_1,\mathbf{b}_1}\left[\exp\left(-\tfrac{1}{2}\mathbf{t}^\top\mathbf{K}_{\mathbf{W}_1,\mathbf{b}_1}(\mathbf{X},\mathbf{X})\mathbf{t}\right)\right] \qquad \text{(char. func. of a Gaussian)}$$

$$\geq \exp\left(-\tfrac{1}{2}\mathbf{t}^\top\mathbb{E}_{\mathbf{W}_1,\mathbf{b}_1}[\mathbf{K}_{\mathbf{W}_1,\mathbf{b}_1}(\mathbf{X},\mathbf{X})]\,\mathbf{t}\right). \qquad \text{(convexity of exp)}$$

This lower bound happens to be the characteristic function of $\mathcal{N}(\mathbf{0}, \mathbb{E}_{\mathbf{W}_1,\mathbf{b}_1}[\mathbf{K}_{\mathbf{W}_1,\mathbf{b}_1}(\mathbf{X},\mathbf{X})])$. Since exp is strictly convex, the characteristic function of $\mathbf{f}_2 \mid \mathbf{X}$ equals the Gaussian lower bound $\forall\mathbf{t}$ if and only if $p(\mathbf{K}_{\mathbf{W}_1,\mathbf{b}_1}(\mathbf{X},\mathbf{X}) \mid \mathbf{W}_1, \mathbf{b}_1) = \mathbb{E}_{\mathbf{f}|\mathbf{X},\mathbf{W}_1,\mathbf{b}_1}[\mathbf{f}_2\mathbf{f}_2^\top]$ is a constant with probability 1. $\qquad\square$

Seeing that $\frac{1}{H_1}\boldsymbol{\sigma}(\mathbf{W}_1^\top\mathbf{x}_i + \beta\mathbf{b}_1)^\top\boldsymbol{\sigma}(\mathbf{W}_1^\top\mathbf{x}_j + \beta\mathbf{b}_1)$ becomes a.s. constant as $H_1 \to \infty$, Lemma 1 re-establishes the result of Neal [69] (see Appx. E.1). Critically, unlike Neal's proof, Lemma 1 neither relies on the central limit theorem nor requires $f_2(\mathbf{f}_1(\cdot))$ to be a neural network; it holds if $p(\mathbf{f}_2 \mid \mathbf{f}_1(\mathbf{x}_1), \ldots, \mathbf{f}_1(\mathbf{x}_N))$ is Gaussian. Therefore, we can generalize it to a larger class of models:

**Lemma 2.** *Let $f_2(\mathbf{f}_1(\cdot)) : \mathbb{R}^D \to \mathbb{R}$ be a hierarchical model where $f_2(\cdot) : \mathbb{R}^{H_1} \to \mathbb{R}$ is a GP and $\mathbf{f}_1(\cdot) : \mathbb{R}^D \to \mathbb{R}^{H_1}$ is a random vector-valued function (including a multilayer hierarchical model). Then $f_2(\mathbf{f}_1(\cdot))$ is a GP if and only if $\mathbb{E}_{\mathbf{f}_2|\mathbf{X},\mathbf{f}_1(\cdot)}[\mathbf{f}_2\mathbf{f}_2^\top] = \mathbb{E}_{\mathbf{f}_2|\mathbf{X}}[\mathbf{f}_2\mathbf{f}_2^\top]$ a.s. for all $\mathbf{X} = [\mathbf{x}_1, \ldots, \mathbf{x}_N]$.*

The covariance perspective from Lemmas 1 and 2 is revealing about GP limits. As $\mathbb{E}_{\mathbf{f}_2|\mathbf{X},\mathbf{f}_1(\cdot)}[\mathbf{f}_2\mathbf{f}_2^\top]$ converges to $\mathbb{E}_{\mathbf{f}_2|\mathbf{X}}[\mathbf{f}_2\mathbf{f}_2^\top]$ the model output becomes less and less dependent on $\mathbf{f}_1(\cdot)$. In other words, $f_2(\mathbf{f}_1(\cdot))$ *loses its hierarchical nature*. We reiterate that Lemma 2 has no requirements about $f_2(\mathbf{f}_1(\cdot))$ transitioning from a finite to infinite basis, nor does it require $f_2(\mathbf{f}_1(\cdot))$ to have additive structure. We demonstrate its generality in the next section with surprising—and troubling—implications.

# 3 Deep Gaussian Processes Collapse to Shallow Gaussian Processes

Deep GP [19, 26, 27, 79] are hierarchical models where layers $\mathbf{f}_1(\cdot) \ldots f_L(\cdot)$ are (vector-valued) GP:

$$\mathrm{DGP}(\mathbf{x}) = f_L \circ \ldots \circ \mathbf{f}_1(\mathbf{x}), \quad \mathbf{f}_i(\cdot) = [f_i^{(1)}(\cdot), \ldots, f_i^{(H_i)}(\cdot)], \quad f_i^{(j)}(\cdot) \overset{\text{i.i.d}}{\sim} \mathcal{GP}[0, k_i(\cdot, \cdot)]. \quad (2)$$

$H_i$ is the width of the $i^{\text{th}}$ GP layer, and the output dimensions of each $\mathbf{f}_i(\cdot)$ are independent. By using GP as the primary hierarchical building blocks, Deep GP are generally nonparametric and, assuming the GP layers use universal kernels [66], have infinite representational capacity (see Appx. B.1).

**Deep GP versus GP.** Deep GP seek to offer more expressivity: conventional single-layer GP—though also nonparametric—are inherently limited by the choice of the prior covariance function [19, 79]. For example, a GP with a RBF covariance is not suitable for data with discontinuities or sharp changes. However, stacking two RBF GP together—$f_2(\mathbf{f}_1(\cdot))$—can overcome this limitation, since $\mathbf{f}_1(\mathbf{x})$ can encode a warping of $\mathbf{x}$ that "smoothes" the input data for $f_2(\cdot)$ (as we will show in Fig. 1). Empirically, Deep GP have been shown to offer much more accurate predictive posteriors than standard GP [e.g. 17, 24, 26, 27, 30, 46, 79].

**Deep GP versus neural networks.** (Bayesian) feed-forward neural networks are a strict subclass of Deep GP, albeit a degenerate one [2, 61, 72]. The first neural network layer is a GP with a linear kernel, while subsequent layers are GP with the kernel $k(\mathbf{z}, \mathbf{z}') = \beta^2 + \frac{1}{H_{i-1}}\sum_{i=1}^{H_{i-1}} \sigma(z_i)\sigma(z_i)$. A neural network, unlike other Deep GP, does not have infinite capacity. Put loosely, a single neural network hidden unit corresponds to a single basis, while in general a single Deep GP unit corresponds to a potentially-infinite basis. See [1, 2, 24, 31, 33, 72] and Appx. B.2 for more discussion on this connection. The critical takeaway is that all of our Deep GP results apply to neural networks as well.

## 3.1 Wide Deep GP are Gaussian Processes

Having established a model where width does not effect capacity, we now establish what remaining effects width has. Empirical evidence suggests that the choice of width impacts Deep GP predictions [19, 46]. In practice it is common to make Deep GP as wide as comparably-sized neural networks; Salimbeni and Deisenroth [79] for example train Deep GP with $\geq 30$ units per layer.

*Surprisingly, here we prove that—in the limit of infinite width—Deep GP collapse to single-layer Gaussian processes.* Our proof relies on the conditional covariance analysis of the previous section. If the GP layers have non-pathological covariance functions[2]—the Deep GP conditional covariance becomes almost surely constant with width (see Lemma 3, Appx. E). Combining this with Lemma 2:

**Theorem 1.** *Let $f_L \circ \ldots \circ \mathbf{f}_1(\mathbf{x})$ be a zero-mean Deep GP (Eq. 2), where each layer satisfies Assumptions 1 and 2 (non-pathological prior covariances that scale with dimensionality—see Appx. E.3). Then $\lim_{H_{L-1}\to\infty} \cdots \lim_{H_1\to\infty} f_L \circ \ldots \circ \mathbf{f}_1(\mathbf{x})$ converges in distribution to a (single-layer) GP.*

---

[2]Any textbook kernel (isotropic kernels, dot product kernels, etc.) or any covariance function with a Fourier-Steiljes representation is "non-pathological;" see Appx. E.3 for formal assumptions.

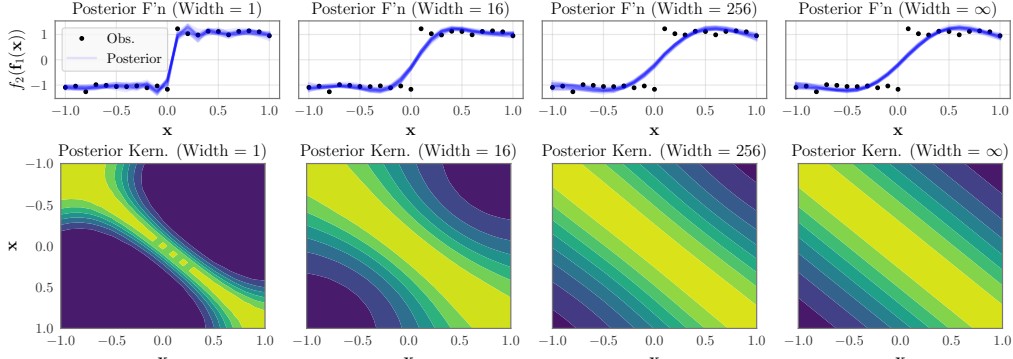

Figure 1: **Top:** Posterior of 2-layer RBF Deep GP fit to a noisy step function. A width-1 Deep GP fits the discontinuity at $\mathbf{x} = 0$. As width increases, the Deep GP converges to a GP with a stationary covariance unable to fit the step. **Bottom:** Average posterior covariance $\mathbb{E}_{\mathbf{f}_1(\mathbf{x}),\mathbf{f}_1(\mathbf{x}')|\mathbf{y}}[k_2(\mathbf{f}_1(\mathbf{x}), \mathbf{f}_1(\mathbf{x}'))]$. The width $= 1$ posterior covariance is non-stationary, with little covariance around $\mathbf{x} = 0$. As width increases, the posterior covariance becomes stationary (as seen by the kernel's constant diagonals).

(See Appx. E for proof.) The implications of Thm. 1 are paradoxical and unsettling. Deep GP are motivated as a more powerful model than standard GP. However, as we make the model wider, we arrive back where we started—a Gaussian process (although one with a different prior covariance).

A neural network gains representational power in its GP limit, transitioning from a finite-basis model to a nonparametric model. The Deep GP limit on the other hand has no additional representational power, since Deep GP are already universal approximators at any width. (Indeed this fact motivates their use as a control.) The only difference between finite and infinite width Deep GP is the prior distribution itself: transitioning from non-Gaussian to Gaussian with increasing width. In the next section, we investigate how this transition affects model performance.

## 4 Large Width Limits the Adaptability of Hierarchical Posteriors

Even with Thm. 1 and its troubling suggestions, it is not immediately clear exactly what is lost in the infinite-width limit. Here, we quantify specific differences in the predictive capabilities of narrow versus wide models. In particular, we analyze Deep GP/neural network posterior distributions, rather than focusing on a single model trained through optimization. We show that these posteriors correspond to a mixture of *data-dependent adaptable bases*; however, as width increases this mixture collapses to the (data-independent) basis of the limiting GP. This result formalizes the often vague notion of *feature learning*, and demonstrates that it is indeed lost in kernel limits.

**Hierarchical posteriors correspond to a data-adaptable bases.** Consider the (finite-width) 2-layer Deep GP $f_2(\mathbf{f}_1(\cdot))$, where $k_1(\cdot, \cdot)$ and $k_2(\cdot, \cdot)$ are the covariance functions of $\mathbf{f}_1(\cdot)$ and $f_2(\cdot)$. Given training data $\mathbf{X}, \mathbf{y}$, define $\mathbf{F}_1 \triangleq [\mathbf{f}_1(\mathbf{x}_1), \ldots, \mathbf{f}_1(\mathbf{x}_N)]$ and $\mathbf{f}_2 \triangleq [f_2(\mathbf{f}_1(\mathbf{x}_1)), \ldots, f_2(\mathbf{f}_1(\mathbf{x}_N))]$. Let $\mathbf{x}^*$ be a test input, and let $\mathbf{f}_1^*$ and $f_2^*$ equal $\mathbf{f}_1(\mathbf{x}^*)$ and $f_2(\mathbf{f}_1(\mathbf{x}^*))$ (see Fig. 5 in Appx. B.3 for a graphical model). Crucially, $\mathbf{f}_2$ and $f_2^*$ only depend on $\mathbf{F}_1$ and $\mathbf{f}_1^*$ through the covariances $\mathbf{K}_2(\mathbf{F}_1, \mathbf{F}_1)$, $\mathbf{k}_2(\mathbf{F}_1, \mathbf{f}_1^*)$, and $k_2(\mathbf{f}_1^*, \mathbf{f}_1^*)$ (which we abbreviate as $\mathbf{K}_2$, $\mathbf{k}_2^*$, and $k_2^{**}$):

$$p(\mathbf{f}_2 \mid \mathbf{K}_2) \sim \mathcal{N}(\mathbf{0}, \mathbf{K}_2), \quad p(f_2^* \mid k_2^{**}, \mathbf{k}_2^*, \mathbf{K}_2, \mathbf{f}_2) \sim \mathcal{N}\left(\mathbf{k}_2^{*\top}\mathbf{K}_2^{-1}\mathbf{f}_2, \; k_2^{**} - \mathbf{k}_2^{*\top}\mathbf{K}_2^{-1}\mathbf{k}_2^*\right),$$

By D-separation [e.g. 16, Ch. 8], we can factorize the posterior distribution as:

$$p(f_2^*, \mathbf{f}_2, \mathbf{K}_2, \mathbf{k}_2^*, k_2^{**} \mid \mathbf{y}) = p(f_2^* \mid \mathbf{f}_2, \mathbf{K}_2, \mathbf{k}_2^*, k_2^{**}) \, p(\mathbf{f}_2 \mid \mathbf{K}_2, \mathbf{y}) \, p(\mathbf{K}_2, \mathbf{k}_2^*, k_2^{**} \mid \mathbf{y}). \quad (3)$$

See derivation in Appx. B.3. Applying the factorization in Eq. (3), the posterior mean is:

$$\mathbb{E}_{f_2^*|\mathbf{y}}[f_2^*] = \mathbb{E}_{\mathbf{K}_2, \mathbf{k}_2^*|\mathbf{y}}\left[\mathbb{E}_{\mathbf{f}_2|\mathbf{K}_2, \mathbf{y}}\left[\mathbf{k}_2^{*\top}\mathbf{K}_2^{-1}\mathbf{f}_2\right]\right] = \mathbb{E}_{\mathbf{K}_2, \mathbf{k}_2^*|\mathbf{y}}\left[\mathbf{k}_2^{*\top}\overbrace{\mathbf{K}_2^{-1}\underbrace{\mathbb{E}_{\mathbf{f}_2|\mathbf{K}_2, \mathbf{y}}[\mathbf{f}_2]}}^{\boldsymbol{\alpha}}\right] \quad (4)$$

$$= \mathbb{E}_{\mathbf{f}_1(\mathbf{x}^*), \mathbf{f}_1(\mathbf{x}_1), \ldots, \mathbf{f}_1(\mathbf{x}_N)|\mathbf{y}}\left[\sum_{i=1}^{N} \alpha_i \, k_2(\mathbf{f}_1(\mathbf{x}_i), \mathbf{f}_1(\mathbf{x}^*))\right], \quad (5)$$

where the second line follows from $\mathbf{K}_2$ and $\mathbf{k}_2^*$ being deterministic given $\mathbf{f}_1(\mathbf{x}^*), \mathbf{f}_1(\mathbf{x}_1), \ldots, \mathbf{f}_1(\mathbf{x}_N)$. The term inside the Eq. (5) expectation is a function from the reproducing kernel Hilbert space (RKHS) defined by $k_2(\mathbf{f}_1(\cdot), \mathbf{f}_1(\cdot))$. We can thus interpret this expectation as an infinite mixture of functions from different Hilbert spaces. Because the mixture distribution $p(\mathbf{f}_1(\mathbf{x}^*), \mathbf{f}_1(\mathbf{x}_1), \ldots, \mathbf{f}_1(\mathbf{x}_N) \mid \mathbf{y})$ depends on $\mathbf{y}$, Eq. (5) is an *adaptive data-dependent mixture of RKHS*.

**Adaptability is lost in the Gaussian process limit.** What happens to Eq. (5) as $f_2(\mathbf{f}_1(\cdot))$ becomes a Gaussian process in the limit of infinite-width? Recall from Lemma 2 that the conditional prior covariance becomes deterministic as $f_2(\mathbf{f}_1(\cdot))$ converges to a GP. In other words, the prior and posterior distributions over $\mathbf{K}_2$ and $\mathbf{k}_2^*$ become atomic: $p(\mathbf{K}_2, \mathbf{k}_2^*) = p(\mathbf{K}_2, \mathbf{k}_2^* \mid \mathbf{y}) = \delta\,[\mathbf{K}_{\text{lim}}, \mathbf{k}_{\text{lim}}^*]$, where $\mathbf{K}_{\text{lim}}$ and $\mathbf{k}_{\text{lim}}^*$ are shorthand for $\mathbb{E}[\mathbf{f}_2 \mathbf{f}_2^\top]$ and $\mathbb{E}[\mathbf{f}_2 f_2^*]$ respectively. Eq. (4) thus collapses to:

$$\lim_{H_1 \to \infty} \mathbb{E}_{f_2^* \mid \mathbf{y}} [f_2^*] = \mathbb{E}_{\delta[\mathbf{K}_{\text{lim}}, \mathbf{k}_{\text{lim}}^*]} \big[\, \mathbf{k}_2^{*\top} \boldsymbol{\alpha} \,\big] = \sum_{i=1}^N \alpha_i \, k_{\text{lim}}(\mathbf{x}_i, \mathbf{x}^*), \tag{6}$$

which is no longer a mixture of functions from different RKHS. It is instead a function from a single RKHS (that of the limiting GP prior).[3] In other words, while Deep GP (and neural networks) perform *kernel learning (or feature learning)* to adapt to training data, this ability is lost with large width.

**Example.** Consider a Deep GP with RBF covariances $k_1(\mathbf{x}, \mathbf{x}') = \exp\left(-\|\mathbf{x} - \mathbf{x}'\|^2/(2D)\right)$ and $k_2(\mathbf{f}_1(\mathbf{x}), \mathbf{f}_1(\mathbf{x}')) = \exp\left(-\|\mathbf{f}_1(\mathbf{x}) - \mathbf{f}_1(\mathbf{x}')\|^2/(2H_1)\right)$. As we show in Appx. G, this Deep GP converges to a GP with $k_{\text{lim}}(\mathbf{x}, \mathbf{x}') = \exp(\exp(-\|\mathbf{x} - \mathbf{x}'\|^2/(2D)) - 1)$. Note that this limiting covariance is *stationary* and is ill-equipped to model the data step in Fig. 1. However, because $\mathbf{f}_1(\cdot)$ is nonlinear, $k_2(\mathbf{f}_1(\mathbf{x}), \mathbf{f}_1(\mathbf{x}'))$ is *nonstationary*. Fig. 1 (top left) shows that the width-1 Deep GP posterior accurately models this data. The posterior covariance $\mathbb{E}_{\mathbf{f}_1(\mathbf{x}), \mathbf{f}_1(\mathbf{x}') \mid \mathbf{y}}[k_2(\mathbf{f}_1(\mathbf{x}), \mathbf{f}_1(\mathbf{x}'))]$ (bottom left) features long-range correlations near $\mathbf{x} = \pm 1$ and short-range correlations near $\mathbf{x} = 0$. As width increases, we lose this nonstationarity and the posterior becomes a worse fit.

## 5 The Difference Between Width and Depth: A Tail Analysis

Our work so far has troubling implications for large width. On the other hand, empirical evidence has shown that depth improves Deep GP performance—as it does for neural nets [e.g. 46, 72, 79] (though pathologies can emerge [29, 33]). Through a novel tail analysis, we show that width makes Deep GP priors more Gaussian, while depth makes them less Gaussian. In other words, *width and depth have opposite effects on Deep GP tails*, results that again also apply to typical neural networks.

**Deep GP/neural networks are sharply peaked and heavy tailed.** The proof technique used in Lemma 1 can be used to similarly bound the moment generating function of Deep GP marginals:

$$\mathbb{E}_{\mathbf{f}_2} \left[ e^{\mathbf{t}^\top \mathbf{f}_2} \right] = \mathbb{E}_{\mathbf{F}_1} \left[ \mathbb{E}_{\mathbf{f}_2 \mid \mathbf{F}_1} \left[ e^{\mathbf{t}^\top \mathbf{f}_2} \right] \right] \geq \exp\left( \frac{1}{2} \mathbf{t}^\top \mathbb{E}_{\mathbf{F}_1} [\mathbf{K}_2(\mathbf{F}_1, \mathbf{F}_1)] \, \mathbf{t} \right) = \mathbb{E}_{\mathbf{g} \sim \mathcal{N}(\mathbf{0}, \mathbf{K}_{\text{lim}})} \left[ e^{\mathbf{t}^\top \mathbf{g}} \right], \tag{7}$$

where $\mathbf{K}_{\text{lim}} = \mathbb{E}_{\mathbf{f}_2}[\mathbf{f}_2 \mathbf{f}_2^\top] = \mathbb{E}_{\mathbf{F}_1}[\mathbf{K}_2(\mathbf{F}_1, \mathbf{F}_1)]$. Generalizing these bounds to deeper models, we have:

**Theorem 2.** *Let $f_L \circ \ldots \circ \mathbf{f}_1(\cdot)$ be a zero-mean Deep GP. Given a finite set of inputs $\mathbf{X} = [\mathbf{x}_1, \ldots, \mathbf{x}_N]$, define $\mathbf{f}_\ell = [(f_\ell \circ \ldots \circ \mathbf{f}_1(\mathbf{x}_1)), \ldots, (f_\ell \circ \ldots \circ \mathbf{f}_1(\mathbf{x}_N))]$ for $\ell \in [1, L]$, and define $\mathbf{K}_{\text{lim}} = \mathbb{E}_{\mathbf{f}_L}[\mathbf{f}_L \mathbf{f}_L^\top]$. Then, $p(\mathbf{f}_L = \mathbf{0}) \geq \mathcal{N}(\mathbf{g} = \mathbf{0}; \mathbf{0}, \mathbf{K}_{\text{lim}})$.*

**Theorem 3.** *Let $\mathbf{t} \in \mathbb{R}^N$. Using the same setup, notation, and assumptions as Thm. 2, the odd moments of $\mathbf{t}^\top \mathbf{f}_L$ are zero and the even moments larger than 2 are super-Gaussian, i.e. $\mathbb{E}_{\mathbf{f}_L}[(\mathbf{t}^\top \mathbf{f}_L)^r] \geq \mathbb{E}_{\mathbf{g} \sim \mathcal{N}(\mathbf{0}, \mathbf{K}_{\text{lim}})}[(\mathbf{t}^\top \mathbf{g})^r]$ for all even $r \geq 4$. Moreover, if $k_L(\cdot, \cdot)$ is bounded almost everywhere, the moment generating function $\mathbb{E}_{\mathbf{f}_L}[\exp(\mathbf{t}^\top \mathbf{f}_L)]$ exists and is similarly super-Gaussian.*

(See Appx. F for proofs.) Thm. 2 states that Deep GP marginals are more sharply peaked than a moment-matched Gaussian, while Thm. 3 states that they are also more heavy tailed.

**Increasing depth leads to sharper peaks and heavier tails.** To understand how depth affects this tail behavior, we examine the Jensen gap in Eq. (7). Consider a 3-layer Deep GP $f_3(\mathbf{f}_2(\mathbf{f}_1(\cdot)))$. If we

---

[3]To rigorously argue that the infinite-width posterior collapses in this way, we can invoke Proposition 1 from Hron et al. [49]. See Appx. B.4 for details.

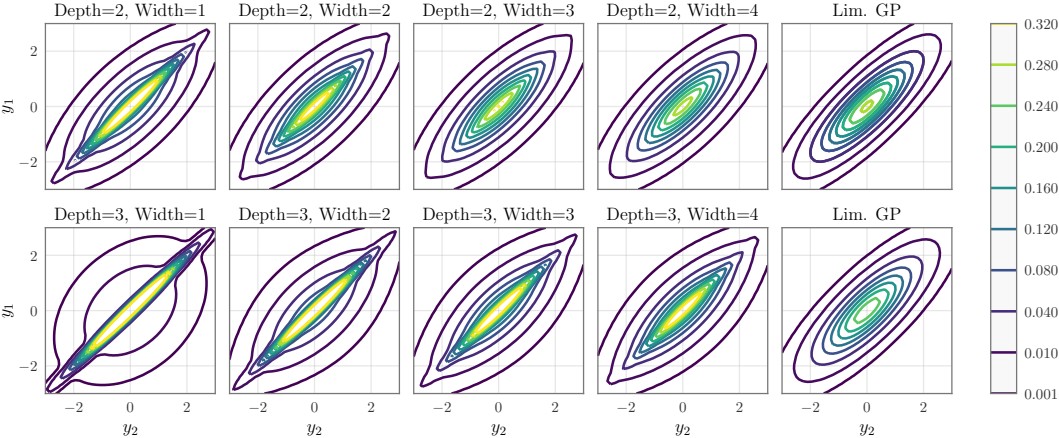

Figure 2: Marginal densities $p(y_1, y_2 \mid \mathbf{x}_1, \mathbf{x}_2)$ for zero-mean Deep GP of various depths and widths on the $N = 2$ dataset $\mathbf{x}_1 = -0.5$, $\mathbf{x}_2 = 0.5$. All 2-layer models have the same second moments (covariance is that of the 3-layer width $= 1$ RBF-RBF-RBF Deep GP). **Left to right:** width increases, marginals become increasingly Gaussian, tails become thinner, and the peak at $[y_1, y_2] = \mathbf{0}$ loses density. **Top to bottom:** depth increases, tails become fatter, and the peak becomes sharper.

extend Eq. (7) to 3-layer models, we see that the Jensen gap cascades:

$$
\underbrace{\mathop{\mathbb{E}}_{\mathbf{F}_1}\left[\mathop{\mathbb{E}}_{\mathbf{F}_2\mid\mathbf{F}_1}\left[\exp\left(\frac{1}{2}\mathbf{t}^\top\mathbf{K}_3\mathbf{t}\right)\right]\right]}_{\text{MGF of 3-layer Deep GP marginal}} \geq \underbrace{\mathop{\mathbb{E}}_{\mathbf{F}_1}\left[\exp\left(\frac{1}{2}\mathbf{t}^\top\mathop{\mathbb{E}}_{\mathbf{F}_2\mid\mathbf{F}_1}[\mathbf{K}_3]\,\mathbf{t}\right)\right]}_{\text{MGF of 2-layer Deep GP marginal}} \geq \underbrace{\exp\left(\frac{1}{2}\mathbf{t}^\top\mathop{\mathbb{E}}_{\mathbf{F}_1}\left[\mathop{\mathbb{E}}_{\mathbf{F}_2\mid\mathbf{F}_1}[\mathbf{K}_3]\right]\mathbf{t}\right)}_{\text{MGF of } \mathcal{N}(\mathbf{0}, \mathbb{E}_{\mathbf{F}_1}[\mathbb{E}_{\mathbf{F}_2\mid\mathbf{F}_1}[\mathbf{K}_3]])},
$$

where $\mathbf{K}_3$ is short for $\mathbf{K}_3(\mathbf{F}_2(\mathbf{F}_1(\mathbf{X})), \mathbf{F}_2(\mathbf{F}_1(\mathbf{X})))$. The middle term is the moment generating function of a 2-layer Deep GP marginal (where the second layer has covariance $\mathbb{E}_{\mathbf{f}_2(\cdot)}[k_3(\mathbf{f}_2(\cdot), \mathbf{f}_2(\cdot))]$). The right-most term is the moment generating function of a (single-layer) Gaussian. Generalizing this cascade, we see that deeper models are more heavy-tailed. A similar analysis on the characteristic function shows that the peak at the prior mean also becomes sharper with depth (see Appx. F).

Adding additional layers to a Deep GP will change the model's prior covariance, and thus the effects of depth cannot solely be explained by a tail analysis [29, 33]. Nevertheless, if we control for this change in covariance, we indeed see that depth leads to heavier tails. In Fig. 2 we compare 2-layer and 3-layer Deep GP. The 3-layer models use GP layers with additively-decomposing RBF covariances, while the 2-layer models use layers constructed to match the 3-layer models' prior covariance (see Appx. H for construction details). The $N = 2$ marginal densities for the 3-layer models (bottom row) are more stretched than the 2-layer densities (top row). We further confirm these effects in Appx. D.

**Increasing width leads to flatter peaks and Gaussian tails.** Conversely, consider what happens when we make the model wider. We define the sequence of increasingly wide 2-layer Deep GP:

$$
\left\{\mathrm{DGP}^{(m)}(\cdot) \triangleq \frac{1}{\sqrt{m}}\sum_{i=1}^{m} f_2^{(i)}(f_1^{(i)}(\cdot))\right\}, \qquad \begin{aligned} f_1^{(i)}(\cdot) &\overset{\text{i.i.d}}{\sim} \mathcal{GP}\left[0, k_1(\cdot, \cdot)\right], \\ f_2^{(i)}(\cdot) &\overset{\text{i.i.d}}{\sim} \mathcal{GP}\left[0, k_2(\cdot, \cdot)\right]. \end{aligned} \tag{8}
$$

$\mathrm{DGP}^{(m)}(\cdot)$ is a width-$m$ Deep GP, where the second layer decomposes additively over the $m$ dimensions. By linearity of expectation, each model in the sequence shares the same prior covariance: $\mathbb{E}[\mathrm{DGP}^{(1)}(\mathbf{x})\,\mathrm{DGP}^{(1)}(\mathbf{x}')] = \mathbb{E}[\mathrm{DGP}^{(2)}(\mathbf{x})\,\mathrm{DGP}^{(2)}(\mathbf{x}')] = \ldots \triangleq k_{\mathrm{lim}}(\mathbf{x}, \mathbf{x}')$. Though each model has the same marginal covariance, the *conditional* covariance $\mathbb{E}_{\mathbf{f}_2\mid\mathbf{F}_1}[\mathbf{f}_2\mathbf{f}_2^\top] = \frac{1}{m}\sum_{i=1}^{m}\mathbf{K}_2(\mathbf{f}_1^{(i)}, \mathbf{f}_1^{(i)})$ becomes increasingly concentrated around $\mathbf{K}_{\mathrm{lim}}(\mathbf{X}, \mathbf{X})$ as $m$ increases. This consequentially shrinks the Jensen gap in Eq. (7), and so the Deep GP marginals become increasingly Gaussian. We again visualize this effect in Fig. 2, which depicts marginal densities from 2-layer and 3-layer Deep GP of various width (see Appx. H for details). Compared with the limiting GP (right), the width-1 densities (left) appear sharper near $[0, 0]$ and more stretched at the tails. As width increases, the peaks and tails look increasingly Gaussian (see also Fig. 7 in Appx. D). In this sense, width has the opposite effect as depth—deeper marginals are less Gaussian, while wider marginals are more Gaussian.

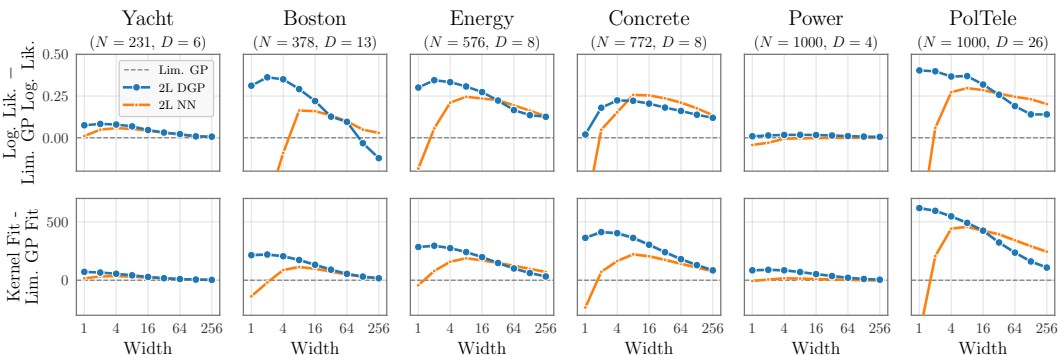

Figure 3: **Top:** Test set log likelihood (LL) of 2-layer Deep GP (and neural networks) regression as a function of width (higher is better). Numbers are shifted so that 0 corresponds to the limiting GP log likelihood. Narrow models achieve the best log likelihood, and performance degrades with width. **Bottom:** Fit of the posterior kernel $k(\mathbf{f}_1(\cdot), \mathbf{f}_1(\cdot))$ on the training data, as measured by Gaussian log marginal likelihood (higher is better). 0 corresponds to the limiting GP log marginal likelihood. Fit becomes increasingly worse with width.

Table 1: Test set log likelihood (LL) of Deep GP regression as a function of depth (higher is better). Depth $= 1$ refers to the limiting GP. For each dataset, the models are constructed to have the same first and second moments. Unlike width, deeper models generally have better performance.

| Depth | Yacht $(N = 231, D = 6)$ | Boston $(N = 378, D = 13)$ | Energy $(N = 576, D = 8)$ | Concrete $(N = 772, D = 8)$ | Power $(N = 1000, D = 4)$ | PolTele $(N = 1000, D = 26)$ |
|---|---|---|---|---|---|---|
| 1 | $-0.532$ | $-0.890$ | $-0.477$ | $-0.663$ | $\mathbf{-0.249}$ | $-0.476$ |
| 2 | $-0.520$ | $-0.684$ | $-0.434$ | $\mathbf{-0.573}$ | $-0.260$ | $-0.381$ |
| 3 | $\mathbf{-0.482}$ | $\mathbf{-0.609}$ | $\mathbf{-0.383}$ | $-0.620$ | $-0.251$ | $\mathbf{-0.318}$ |

# 6 Experiments

## 6.1 Regression with Deep GP and Bayesian Neural Networks

To isolate the effects of width and depth, each experiment compares Deep GP/Bayesian neural networks that share the same first and second prior moments, and the Deep GP models use GP layers with universal kernels. To remove any potential side effects from approximate inference methods, we sample Deep GP/neural network posteriors using NUTS [48] and do not use any stochastic inducing point [46, 79] or finite basis [24] approximations. This inference is costly and scales cubically with $N$; therefore, we subsample all training datasets to $N \leq 1000$. See Appx. H for experimental details.

**Effect of width.** We compare 2-layer Deep GP of various width on 6 regression datasets from the UCI dataset repository [10] (see Appx. D for 3-layer results). The first GP layers use a RBF kernel for the prior covariance, while the second layers use a sum of one-dimensional RBF covariance functions. We additionally compare against the limiting (single-layer) GP with the same prior covariance (**Lim. GP**). For each dataset, we choose hyperparameters that maximize the Lim. GP log marginal likelihood. In Fig. 3 (top row) we see a near-monotonic performance degradation as width increases. The width $= 2$ optimum may represent the "sweet spot" for Deep GP width, but it may instead be a side-effect of inference difficulties for width $= 1$ models (see Appx. D for a control experiment). Regardless, as our theory predicts, *width is detrimental to Deep GP predictive performance*.

We repeat the experiment for 2-layer neural networks (and 3-layer models in Appx. D), where here the Lim. GP corresponds to the arc-cosine kernel [23, 56]. Fig. 3 indicates an optimal width with regards to test set log likelihood, usually between $8 - 16$ hidden units. We expect this optimum exists (and differs from the Deep GP optimum) because narrow models have too few basis functions for these datasets. Nevertheless, after sufficient capacity, *width is harmful to Bayesian neural networks*.

**Adaptable versus non-adaptable RKHS.** One way to measure the "fit" of a kernel $k(\cdot, \cdot)$ on a regression training dataset $\mathbf{X}, \mathbf{y}$ is the Gaussian log marginal likelihood $\log \mathcal{N}(\mathbf{y}; \mathbf{0}, \mathbf{K}(\mathbf{X}, \mathbf{X}) + \sigma^2 \mathbf{I})$, where $\sigma^2$ is an observational noise parameter [e.g. 77]. To demonstrate how Deep GP/neural network

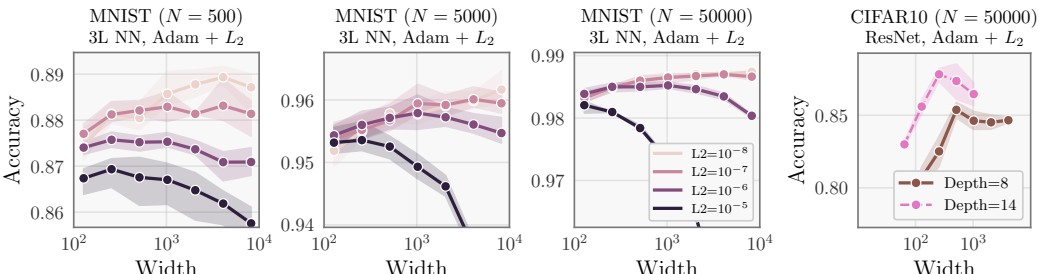

Figure 4: Effect of width on standard (non-Bayesian) neural networks. Shaded regions depict standard error. **Left:** 3-layer MLP trained on subsets of MNIST. With large values of L2 regularization, model performance is maximized when width $\leq 1,000$. For small values of L2 regularization (e.g. $10^{-8}$, which corresponds to a prior of $\mathcal{N}(0, 20,000)$ on the parameters), there is little accuracy loss with increasing width. It is possible that our theory does not apply to models with little L2 regularization which have little Bayesian interpretation. **Right:** Wide ResNet models (8-layer and 14-layer variants) trained on CIFAR-10. For both depths, accuracy is optimal when width $\leq 500$.

posteriors correspond to adaptable RKHS mixtures, the bottom row of Fig. 3 plots the "kernel fit" of $k_2(\mathbf{f}_1(\cdot), \mathbf{f}_1(\cdot))$ for posterior samples of $\mathbf{f}_1(\cdot)$ (see Eq. 5). A higher fit corresponds to a model that is better adapted to the dataset $\mathbf{X}, \mathbf{y}$. We see that narrower Deep GP almost universally achieve better kernel fit than wider Deep GP, which converge to the same fit as the limiting GP. (Standard deviations, depicted by shaded regions, are generally imperceptible.) Bayesian neural networks achieve best "kernel fit" at $8 - 16$ hidden units, and then converge to the limiting Deep GP with further width.

**Effect of depth, controlling for covariance.** Table 1 displays Deep GP test set log likelihood as a function of depth. Again, we isolate the tail effects of depth by ensuring that all models share the same first and second moments. We construct a GP and a 2-layer Deep GP that match the moments of a 3-layer width $= 2$ Deep GP with RBF covariances, and we use hyperparameters that maximize the limiting GP marginal likelihood for each dataset. Note that computing the limiting covariance of $\geq 3$ layer models involves intractable integrals that we approximate with quadrature (see Appx. G). Our findings confirm that—in this controlled setting—depth unlike width improves test set performance.

### 6.2 Standard (Optimized, Non-Bayesian) Neural Networks

We now turn to standard (optimized, non-Bayesian) neural networks. While our theoretical results primarily apply to full posteriors over models, our goal is to see if our theory can also be predictive in "real world" neural networks without a Bayesian treatment. There is reason to believe that our theory should be applicable in these settings, since standard neural network training with L2 regularization is equivalent to maximum a posteriori inference with Gaussian priors. To that end, we ensure some correspondence between these experiments and our Bayesian experiments. In particular, we measure the effects of width on networks with fixed values of L2 regularization,[4] which corresponds to a fixed prior on neural network parameters. Additionally, models are trained without data augmentation, as data augmentation does not have a probabilistic interpretation [51].

Fig. 4 (left) depicts test set accuracy for increasingly wide models trained on MNIST [55]. Each network is a MLP with 3 layers (i.e. 2 hidden layers). Following the GP-limiting neural network construction in Eq. (1), we scale the outputs of layer $\ell$ by $1/\sqrt{H_{\ell-1}}$. We measure the effect of width over networks with various L2 regularization constants ($10^{-5}$, $10^{-6}$, $10^{-7}$, and $10^{-8}$) which respectively correspond to priors of $\mathcal{N}(0, 2)$, $\mathcal{N}(0, 20)$, $\mathcal{N}(0, 200)$, and $\mathcal{N}(0, 2000)$ when $N = 50,000$. We train these sequences on various-sized subsets of the training data ($N = 500$, $N = 5,000$, and $N = 50,000$). From this figure we can observe several phenomena. For larger values of L2 regularization, we see a distinct maximum in accuracy, typically around width $\approx 1,000$. For smaller values of L2 regularization, wider models tend to perform better (and indeed, for this dataset/model combination it appears that less regularization tends to be beneficial to overall performance). We would note that these low regularization constants correspond to arguably unreasonable parametric

---

[4]In other words, we do not consider the regularization constant to be a hyperparameter that we optimize over for the purposes of these experiments.

priors like $\mathcal{N}(0, 2000)$, and so a Bayesian interpretation of these models may not be applicable. In such settings, it is more likely that the interpolation analysis of Belkin et al. [12] is a better model of performance, since this analysis explicitly focuses on the low-regularization setting.

Fig. 4 (right) depicts 8- and 14-layer ResNets [47] trained on the CIFAR-10 dataset [54]. We use the hyperparameters from the original ResNet paper, which have been shown to be efficacious on both narrow and wide variants of ResNet models [92]. (This training procedure uses a L2 coefficient of $10^{-4}$, which corresponds to a prior of $\mathcal{N}(0, 0.2)$ for each parameter when $N = 50{,}000$.) For both depths, we observe that performance is optimal when width is between $500$ and $1{,}000$. While it is possible that different hyperparameters may yield different outcomes, these results indeed suggest that *large width can adversely affect standard neural networks* once sufficient capacity is reached.

# 7    Discussion

This paper shows that, across typical neural networks (with L2 regularization), Deep GP, and Bayesian neural networks, *large width can be detrimental to model performance*.

Even with these results, we can ask when width might be desirable? First, we note that our results analyze exact posteriors or MAP solutions, and does not focus on practical considerations with regards to obtaining these solutions. We do not consider the effect that width might have on approximate inference methods, which are commonly used with Bayesian neural networks and Deep GP in practice [e.g. 18, 34, 79]. For conventional neural networks, poor conditioning and non-convexity make it challenging to obtain a MAP solution. The optimization dynamics—which depend on numerous factors like learning rates, initializations, and choice of optimizer—may be improved by width, as wider models tend to have more favorable optimization landscapes [7, 28, 59, 70, 82]. Consequentially, wider models may obtain better performance due to these practical considerations.

Secondly—as noted in Sec. 6.2—while we notice detrimental effects of width on neural networks with a Bayesian interpretation (i.e. inferring a parameter posterior or optimizing parameters with L2 regularization), we do not see these effects when such an interpretation does not exist (i.e. optimizing parameters with almost no L2 regularization). Our theoretical findings assume that layers are conditionally Gaussian, and different priors may have different effects. We note that much of the preliminary works on NTK assume no explicit regularization during training [28, 52, 57] (with the notable exception of Wei et al. [86]), and so our findings may be at odds with the empirical findings around these models [e.g. 9, 38, 39]. Moreover, recent work has proposed (non-Bayesian) infinite-width constructions that avoid any limiting kernel behavior [e.g. 22, 43, 65, 90], and so our findings would not apply to these models. We emphasize that our results do not conflict with these prior works, but rather reflect a different perspective. The models we study correspond to a Gaussian prior on parameters, and so relaxing this correspondence may lessen the consequences of width that we observe. Nevertheless, our results suggest that the inductive bias of width may be harmful, even if these undesirable effects can be avoided via careful construction.

Finally, it is worth considering when one might still choose a conventional shallow GP over a deep model. An often-touted benefit of Gaussian processes is the ability to encode prior domain knowledge via the choice of covariance function. In Appx. C, we prove that certain prior covariances cannot be expressed by adaptable hierarchical models. For example, a Deep GP that is composed of stationary GP layers cannot model anti-correlations a priori (Thm. 4, Appx. C), whereas (single-layer) stationary GP can have positive and negative prior covariances. Nevertheless, Deep GP are capable of modeling many common covariance functions, including the RBF, Matérn, and rational quadratic kernels. In Appx. C we demonstrate a 2-layer Deep GP construction of any width that is capable of producing prior covariances that match most isotropic kernels (Thm. 5, Appx. C). In other words, a Deep GP can match the first and second moments of most GP, while also offering an adaptable posterior.

## Acknowledgments and Disclosure of Funding

We would like to thank Elliott Gordon-Rodriguez for his help with the proofs. This work was supported by the Simons Foundation, McKnight Foundation, the Grossman Center, and the Gatsby Charitable Trust.

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
