# Supplementary Information for: The Limitations of Large Width in Neural Networks: A Deep Gaussian Process Perspective

**Geoff Pleiss**
Columbia University
gmp2162@columbia.edu

**John P. Cunningham**
Columbia University
jpc2181@columbia.edu

## A  Broader Impact

This paper analyzes two existing classes of models: Deep GP and neural networks. We believe that our findings will be of interest to researchers and machine learning practitioners, offering useful guidance for Deep GP and neural network architectures. Because we are neither introducing new algorithms nor introducing new use cases of existing algorithms, we do not foresee any major ethical impacts from this work. However, we do note that this paper primarily focuses on how width affects performance metrics (e.g. accuracy, log likelihood, etc.) and does not focus on other metrics that may be of interest to practitioners and society at large (e.g. interpretability, energy usage, fairness, etc.).

## B  Deep Gaussian Processes

In this section we discuss various Deep GP facts presented throughout the main paper.

### B.1  Capacity of Deep GP

Here we formalize the claim that Deep GP have "infinite capacity." Standard Gaussian processes are nonparametric, and—if the prior covariance is a universal kernel [66]—then any function (or an arbitrarily precise approximation thereof) is a draw from its prior. A Deep GP composes multiple GP as different layers. If all GP layers use universal kernels for covariance priors, then any (arbitrarily precise approximation of a) function $h(\cdot)$ is a draw from the Deep GP prior. (Draw the identity function from the first $L - 1$ GP layers, and then draw $h(\cdot)$ from the last GP layer.) In this sense, Deep GP as well as standard GP can model any function to arbitrary precision and in this sense have infinite capacity.

### B.2  Connection Between Neural Networks and Deep GP

Throughout this paper we note that feed-forward (Bayesian) neural networks are a degenerate subclass of Deep Gaussian processes. We will now formalize this connection, which has also been noted in several previous works [e.g. 2, 61, 72].

To show that the neural network defined in Eq. (1) is a (degenerate) Deep GP, we must show that each of its layers corresponds to a (degenerate, vector-valued) Gaussian process. Recall that a Gaussian process $g(\cdot) \sim \mathcal{GP}$ is a distribution over functions where every finite marginal distribution $\mathbf{f} = [g(\mathbf{x}_1), \ldots, g(\mathbf{x}_N)]$ is multivariate Gaussian consistent with some covariance function. The first layer of the Eq. (1) neural network is given by

$$f_1^{(i)}(\mathbf{x}) = \mathbf{w}_1^{(i)\top}(\mathbf{x}) + \beta b_1^{(i)},$$

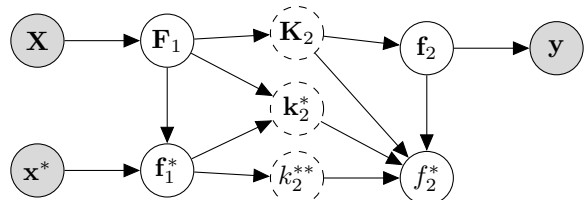

Figure 5: 2-layer Deep GP. $\mathbf{X}$, $\mathbf{y}$ are the training data; $\mathbf{x}^*$ is some unobserved test input. $\mathbf{F}_1$ and $\mathbf{f}_1^*$ are the first layer outputs for the training inputs and test input, respectively. $\mathbf{f}_2$ and $f_2^*$ are the second layer outputs for the train/test inputs, which only depend on $\mathbf{F}_1$, $\mathbf{f}_1^*$ through the prior covariance matrices $\mathbf{K}_2 = \mathbf{K}_2(\mathbf{F}_1, \mathbf{F}_1)$, $\mathbf{k}_2^* = \mathbf{k}_2(\mathbf{F}_1, \mathbf{f}_1^*)$, and $k_2^{**} = k_2(\mathbf{f}_1^*, \mathbf{f}_1^*)$.

Thus, the first layer corresponds to a vector-valued Gaussian process with the (degenerate) linear prior covariance. The second layer of the Eq. (1) neural network is given by

$$f_2(\mathbf{z}) = \tfrac{1}{\sqrt{H_1}} \mathbf{w}_2^\top \boldsymbol{\sigma}(\mathbf{z}) + \beta b_2,$$

where again the entries of $\mathbf{w}_2$ and $b_2$ are i.i.d. unit Normal. We have that $\mathbf{f}_2 \mid \mathbf{F}_1 = \mathcal{N}(\mathbf{0}, \beta + \boldsymbol{\sigma}(\mathbf{F}_1)\boldsymbol{\sigma}(\mathbf{F}_1)^\top)$, where $\mathbf{F}_1 = [\mathbf{f}_1(\mathbf{x}_1), \dots, \mathbf{f}_1(\mathbf{x}_N)]$ and $\boldsymbol{\sigma}(\mathbf{F}_1)$ corresponds to the elementwise nonlinearity $\boldsymbol{\sigma}(\cdot)$ applied to each entry of $\mathbf{F}_1$. Thus, the second layer also corresponds to a Gaussian process with a degenerate prior covariance.

**Neural networks versus Deep GP.** While (Bayesian) neural networks meet the definition of a Deep GP, their covariance functions only correspond to a finite basis and therefore they do not have the same properties as nonparametric Deep GP (i.e. the ability to model any function to arbitrary precision). In this sense, it is common to treat neural networks and Deep GP as two separate classes of models with different predictive properties. However, we emphasize that the theoretical results in this paper make no assumptions about whether or not a Deep GP is nonparametric, and therefore the behaviors that we analyze are inherent to both classes of models. In this sense, it is useful for our purposes to group nonparametric Deep GP and neural networks into a single class of models.

We also note that Deep GP and (Bayesian) neural networks can both be generalized to other hierarchical models, such as Deep Kernel Processes [3].

### B.3 Factorization of Deep GP Posterior

Here we supply additional details for the Sec. 4 derivation of the Deep GP posterior mean. Consider a 2-layer zero-mean Deep-GP $f_2(\mathbf{f}_1(\cdot))$. Fig. 5 depicts the relationships between these variables, using the same notation as in Sec. 4. Now consider the posterior distribution

$$p(f_2^*, \mathbf{f}_2, \mathbf{F}_1, \mathbf{f}_1 \mid \mathbf{y}) \;=\; p(f_2^* \mid \mathbf{f}_2, \mathbf{f}_1^*, \mathbf{F}_1, \mathbf{y})\, p(\mathbf{f}_2 \mid \mathbf{F}_1, \mathbf{y}, \mathbf{f}_1^*)\, p(\mathbf{f}_1^*, \mathbf{F}_1 \mid \mathbf{y}),$$

where we have omitted the dependence on $\mathbf{X}$ and $\mathbf{x}^*$ for clarity. Apply the rules of D-separation using Fig. 5, we see that $f_2^*$ only depends on $\mathbf{y}$ through $\mathbf{f}_2$, and thus $f_2^*$ is conditionally independent from $\mathbf{y}$ given $\mathbf{f}_2$. Furthermore, we see that $\mathbf{f}_2$ is only connected to $\mathbf{f}_1^*$ through $f_2^*$, and so $\mathbf{f}_2$ is conditionally independent from $\mathbf{f}_1^*$ if $f_2^*$ is marginalized out. Thus, we can simplify the posterior factorization to:

$$p(f_2^*, \mathbf{f}_2, \mathbf{F}_1, \mathbf{f}_1 \mid \mathbf{y}) \;=\; p(f_2^* \mid \mathbf{f}_2, \mathbf{f}_1^*, \mathbf{F}_1)\, p(\mathbf{f}_2 \mid \mathbf{F}_1, \mathbf{y})\, p(\mathbf{f}_1^*, \mathbf{F}_1 \mid \mathbf{y}), \qquad (9)$$

Crucially, $\mathbf{f}_2$ and $f_2^*$ only depend on $\mathbf{F}_1$ and $\mathbf{f}_1^*$ through $\mathbf{K}_2$, $\mathbf{k}_2^*$, and $k_2^{**}$:

$$p(\mathbf{f}_2 \mid \mathbf{K}_2) \sim \mathcal{N}(\mathbf{0}, \mathbf{K}_2), \quad p(f_2^* \mid k_2^{**}, \mathbf{k}_2^*, \mathbf{K}_2, \mathbf{f}_2) \sim \mathcal{N}\left(\mathbf{k}_2^{*\top} \mathbf{K}_2^{-1} \mathbf{f}_2,\; k_2^{**} - \mathbf{k}_2^{*\top} \mathbf{K}_2^{-1} \mathbf{k}_2^*\right),$$

(If $f_2(\mathbf{f}_1(\cdot))$ is a neural network or any other degenerate Deep GP, the $\mathbf{K}_2^{-1}$ term can be replaced with its pseudoinverse.) This relationship is also depicted graphically in Fig. 5. $\mathbf{K}_2$, $\mathbf{k}_2^*$, and $k_2^{**}$ are deterministic given $\mathbf{F}_1$ and $\mathbf{f}_1^*$, and we do not ultimately care about the values of $\mathbf{F}_1$ and $\mathbf{f}_1^*$ since they are intermediate latent variables. Therefore, we can rewrite the factorization in Eq. (9) where we replace $\mathbf{F}_1$, $\mathbf{f}_1^*$ with $\mathbf{K}_2$, $\mathbf{k}_2^*$, and $k_2^{**}$:

$$p(f_2^*, \mathbf{f}_2, \mathbf{K}_2, \mathbf{k}_2^*, k_2^{**} \mid \mathbf{y}) \;=\; p(f_2^* \mid \mathbf{f}_2, \mathbf{K}_2, \mathbf{k}_2^*, k_2^{**})\, p(\mathbf{f}_2 \mid \mathbf{K}_2, \mathbf{y})\, p(\mathbf{K}_2, \mathbf{k}_2^*, k_2^{**} \mid \mathbf{y}). \qquad (10)$$

Applying the factorization in Eq. (10), the posterior mean is:

$$\mathbb{E}_{f_2^*|\mathbf{y}} \left[ f_2^* \right] = \mathbb{E}_{\mathbf{K}_2,\mathbf{k}_2^*,k_2^{**}|\mathbf{y}} \left[ \mathbb{E}_{\mathbf{f}_2|\mathbf{K}_2,\mathbf{y}} \left[ \mathbb{E}_{f_2^*|\mathbf{f}_2,\mathbf{K}_2,\mathbf{k}_2^*,k_2^{**}} \left[ f_2^* \right] \right] \right]$$

$$= \mathbb{E}_{\mathbf{K}_2,\mathbf{k}_2^*|\mathbf{y}} \left[ \mathbb{E}_{\mathbf{f}_2|\mathbf{K}_2,\mathbf{y}} \left[ \mathbf{k}_2^{*\top} \mathbf{K}_2^{-1} \mathbf{f}_2 \right] \right] \tag{11}$$

$$= \mathbb{E}_{\mathbf{K}_2,\mathbf{k}_2^*|\mathbf{y}} \left[ \mathbf{k}_2^{*\top} \mathbf{K}_2^{-1} \overbrace{\mathbb{E}_{\mathbf{f}_2|\mathbf{K}_2,\mathbf{y}} \left[ \mathbf{f}_2 \right]}^{\boldsymbol{\alpha}} \right] \tag{12}$$

(Again, if $f_2(\mathbf{f}_1(\cdot))$ is a neural network, the $\mathbf{K}_2^{-1}$ term in Eqs. (11) and (12) can be replaced with its pseudoinverse.) Finally, since $\mathbf{K}_2, \mathbf{k}_2^*$ are deterministic transforms of $\mathbf{f}_1(\mathbf{x}^*), \mathbf{f}_1(\mathbf{x}_1), \ldots, \mathbf{f}_1(\mathbf{x}_N)$, we can rewrite Eq. (12) as:

$$\mathbb{E}_{f_2^*|\mathbf{y}} \left[ f_2^* \right] = \mathbb{E}_{\mathbf{f}_1(\mathbf{x}^*),\mathbf{f}_1(\mathbf{x}_1),\ldots,\mathbf{f}_1(\mathbf{x}_N)|\mathbf{y}} \left[ \sum_{i=1}^{N} \alpha_i \, k_2(\mathbf{f}_1(\mathbf{x}_i), \mathbf{f}_1(\mathbf{x}^*)) \right],$$

which completes the derivation of Eq. (5) in Sec. 4.

### B.4    A Rigorous Argument for Deep GP Posterior Collapse in the Infinite-Width Limit

In Sec. 4 (Eq. 6), we argue that the posterior mean of an infinite-width Deep GP collapses to the posterior of the limiting GP. To make this argument mathematically rigorous (and to demonstrate that this limiting posterior does not "blow up"), we need to establish that convergence to a GP prior also implies convergence to the corresponding GP posterior. Hron et al. [49, Proposition 1] proves that this is indeed the case, with only mild assumptions on the likelihood:

**Proposition 1 of Hron et al. [49].** *Assume $P_{f_n} \Rightarrow P_f$ (where $P_{f_n}$ in some sequence of priors, $P_f$ is some limiting prior, and $\Rightarrow$ denotes convergence in distribution) on the usual Borel product $\sigma$-algebra, Assumption 1 from Hron et al. [49] holds for the chosen likelihood $\ell$, and that $\int \ell \, dP_f > 0$. Then,*

$$P_{f_n|D} \Rightarrow P_{f|D}$$

*with $P_{f_n|D}$ and $P_{f|D}$ the Bayesian posteriors induced by the likelihood $\ell$ and respectively the priors $P_{f_n}$ and $P_f$.*

Common likelihoods, such as the Gaussian likelihood for regression or the categorical likelihood for multiclass classification, satisfy the assumptions of this proposition.

### B.5    Formalizing the Notion of "Sufficient Capacity" for Neural Networks

Throughout the paper, we argue that width harms model performance once a model has "sufficient capacity" for a given dataset. We intentionally keep this notion vague, since there are various measures of capacity that—while useful for analyzing trends in network architectures—are an imperfect quantification of the power of a neural network. Nevertheless, under any standard definition of capacity, such as VC dimension, neural networks have finite capacity whereas nonparametric Deep GP with universal covariance functions have infinite capacity. Our theory in Secs. 3 to 5 suggests that width controls a capacity/adaptability trade-off for parametric neural networks, analogous to other classic machine learning trade-offs. While the optimal capacity of a neural network depends on several hard-to-measure factors and dataset-dependent features, it stands to reason that—after sufficient width—additional neurons make the prior distribution increasingly Gaussian while offering little additional gains in modeling precision (as suggested by the orange lines in Figs. 3 and 9). This is the regime that we refer to as "sufficient capacity."

## C    What Prior Covariance Functions can be Modeled by Deep Gaussian Processes?

The functional properties of standard Gaussian processes are largely determined by the choice of prior covariance function [77]. Any positive definite function is a valid GP covariance, making it possible to encode many types of functional priors. For Deep GP, it is reasonable to assume that its

prior second moment also has significant influence on its inductive bias and functional properties. To that end, it is of interest to determine what covariances can be modeled by Deep GP a priori.

We present two theoretical results in this section. The first is a negative result (Thm. 4), which states that Deep GP with stationary GP layers can only model non-negative covariance functions a priori. This is in contrast to standard GP, which are capable of expressing anti-correlations with stationary covariance priors. The second is a positive result (Thm. 5), which demonstrates a Deep GP construction capable of modeling most isotropic covariance priors. We note that isotropic functions (e.g. RBF, Matérn, rational quadratic, etc.) are some of the most common covariance priors.

**Theorem 4.** *Let* $\mathrm{DGP}(\cdot) = f_L \circ \cdots \circ \mathbf{f}_1(\cdot)$ *be a L-layer zero-mean Deep GP where* $f_L(\cdot)$ *has a mean-square continuous stationary prior covariance. Then* $\mathbb{E}[\mathrm{DGP}(\mathbf{x})\,\mathrm{DGP}(\mathbf{x}')] > 0$ *for all* $\mathbf{x}, \mathbf{x}'$.

*Proof of Theorem 4.* Throughout the proof, we will use the shorthand $\mathbf{f}_\ell = \mathbf{f}_\ell \circ \ldots \circ \mathbf{f}_1(\mathbf{x})$ and $\mathbf{f}'_\ell = \mathbf{f}_\ell \circ \ldots \circ \mathbf{f}_1(\mathbf{x}')$. Because $k_L(\cdot, \cdot)$ is stationary, we can express $\mathbb{E}[\mathrm{DGP}(\mathbf{x})\,\mathrm{DGP}(\mathbf{x}')]$ as:

$$\mathbb{E}\left[\mathrm{DGP}(\mathbf{x})\,\mathrm{DGP}(\mathbf{x}')\right] = \int k_L(\mathbf{f}_{L-1} - \mathbf{f}'_{L-1})\,\mathrm{d}p(\mathbf{f}_{L-1}, \mathbf{f}'_{L-1}).$$

Moreover, by Bochner's theorem, we can express $k_2(\mathbf{f}_{L-1} - \mathbf{f}'_{L-1})$ as the Fourier transform of some positive finite measure $\mu(\boldsymbol{\xi})$:

$$\mathbb{E}\left[\mathrm{DGP}(\mathbf{x})\,\mathrm{DGP}(\mathbf{x}')\right] = \int \left( \int \exp(i\,\boldsymbol{\xi}^\top(\mathbf{f}_{L-1} - \mathbf{f}'_{L-1}))\,\mathrm{d}\mu(\boldsymbol{\xi}) \right)\,\mathrm{d}p(\mathbf{f}_{L-1}, \mathbf{f}'_{L-1}). \quad (13)$$

Note that $|\exp(i\,\cdot)|$ is bounded everywhere, and $\mu(\boldsymbol{\xi})$ and $p(\mathbf{f}_{L-1}, \mathbf{f}'_{L-1})$ are finite measures. Therefore we can switch the order of integration in Eq. (13):

$$\mathbb{E}\left[\mathrm{DGP}(\mathbf{x})\,\mathrm{DGP}(\mathbf{x}')\right] = \int \left( \int \exp(i\,\boldsymbol{\xi}^\top(\mathbf{f}_{L-1} - \mathbf{f}'_{L-1})\,\mathrm{d}p(\mathbf{f}_{L-1}, \mathbf{f}'_{L-1}) \right)\,\mathrm{d}\mu(\boldsymbol{\xi}). \quad (14)$$

$$= \int \left( \int \exp \left( i \begin{bmatrix} \boldsymbol{\xi} \\ -\boldsymbol{\xi} \end{bmatrix}^\top \begin{bmatrix} \mathbf{f}_{L-1} \\ \mathbf{f}'_{L-1} \end{bmatrix} \right)\,\mathrm{d}p \left( \begin{bmatrix} \mathbf{f}_{L-1} \\ \mathbf{f}'_{L-1} \end{bmatrix} \right) \right)\,\mathrm{d}\mu(\boldsymbol{\xi}).$$

Applying the characteristic function lower bound for Deep GP marginals (see Appx. F, Eq. 33):

$$\mathbb{E}\left[\mathrm{DGP}(\mathbf{x})\,\mathrm{DGP}(\mathbf{x}')\right] \geq \int \exp \left( -\frac{1}{2} \begin{bmatrix} \boldsymbol{\xi} \\ -\boldsymbol{\xi} \end{bmatrix}^\top \mathbb{E}\left[ \begin{bmatrix} \mathbf{f}_{L-1} \\ \mathbf{f}'_{L-1} \end{bmatrix} \begin{bmatrix} \mathbf{f}_{L-1} \\ \mathbf{f}'_{L-1} \end{bmatrix}^\top \right] \begin{bmatrix} \boldsymbol{\xi} \\ -\boldsymbol{\xi} \end{bmatrix} \right)\,\mathrm{d}\mu(\boldsymbol{\xi}). \quad (15)$$

The integrand in Eq. (15) is a real-valued exponential, and so it is strictly positive. Since $\mu(\boldsymbol{\xi})$ is a positive measure, we have that $\mathbb{E}[\mathrm{DGP}(\mathbf{x})\,\mathrm{DGP}(\mathbf{x}')] > 0$. $\qquad \square$

**Theorem 5.** *Let* $k_{lim}(\mathbf{x}, \mathbf{x}') = \varphi(\|\mathbf{x} - \mathbf{x}'\|_2^2)$ *be a mean-square continuous isotropic covariance function that is valid on* $\mathbb{R}^D \times \mathbb{R}^D$ *for all* $D \in \mathbb{N}$. *For any width* $H_1 \in \mathbb{N}$, *the exists a 2-layer Deep GP* $f_2(\mathbf{f}_1(\cdot))$ *with* $\mathbf{f}_1(\cdot) : \mathbb{R}^D \to \mathbb{R}^{H_1}$ *and* $f_2(\cdot) : \mathbb{R}^{H_1} \to \mathbb{R}$ *where* $\mathbb{E}[f_2(\mathbf{f}_1(\mathbf{x}))f_2(\mathbf{f}_1(\mathbf{x}'))] = k_{lim}(\mathbf{x}, \mathbf{x}')$.

*Proof of Theorem 5.* A classic result from Schoenberg [80, Thm. 2] is that, for any mean-square continuous isotropic covariance function $\varphi(\|\mathbf{x} - \mathbf{x}'\|_2^2)$ that is valid on $\mathbb{R}^D \times \mathbb{R}^D$ for all $D \in \mathbb{N}$, there exists some positive finite measure $\mu(\beta)$ such that

$$\varphi(\|\mathbf{x} - \mathbf{x}'\|_2^2) = \int \exp \left( -\frac{1}{2} \|\mathbf{x} - \mathbf{x}'\|_2^2\,\beta \right)\,\mathrm{d}\mu(\beta). \quad (16)$$

Let $\mathrm{DGP}^{(1)}(\cdot) = f_2(f_1(\cdot))$ be a 2-layer zero-mean Deep GP with width $H_1 = 1$, and let $k_1(\mathbf{x}, \mathbf{x}') = \mathbf{x}^\top \mathbf{x}'$ and $k_2(z, z') = \int \exp(i\,\beta(z - z'))\,\mathrm{d}\mu(\beta)$. (By Bochner's theorem, we know that $k_2(\cdot, \cdot)$ is a valid covariance function.) Define $\tau = f_1(\mathbf{x}) - f_1(\mathbf{x}')$. Since $f_1(\mathbf{x})$ and $f_1(\mathbf{x}')$ are jointly Gaussian:

$$p \left( \begin{bmatrix} f_1(\mathbf{x}) \\ f_1(\mathbf{x}') \end{bmatrix} \right) = \mathcal{N} \left( \begin{bmatrix} 0 \\ 0 \end{bmatrix}, \begin{bmatrix} \mathbf{x}^\top \mathbf{x} & \mathbf{x}^\top \mathbf{x}' \\ \mathbf{x}'^\top \mathbf{x} & \mathbf{x}'^\top \mathbf{x}' \end{bmatrix} \right)$$

we have that $p(\tau) = \mathcal{N}(\mathbf{0}, \|\mathbf{x} - \mathbf{x}'\|_2^2)$. Substituting $\tau$ into Eq. (14), we have:

$$\mathbb{E}\left[\mathrm{DGP}^{(1)}(\mathbf{x})\,\mathrm{DGP}^{(1)}(\mathbf{x}')\right] = \int \left(\int \exp(i\,\beta\tau)\,\mathrm{d}p(\tau)\right)\,\mathrm{d}\mu(\beta).$$

$$= \int \exp\left(-\frac{1}{2}\|\mathbf{x} - \mathbf{x}'\|_2^2\,\beta\right)\,\mathrm{d}\mu(\beta). \tag{17}$$

Thus we have a width-1 Deep GP with prior covariance $k_{\mathrm{lim}}(\cdot, \cdot)$. We can extend this construction to 2-layer Deep GP of any width using the additive sequence defined in Eq. (8). $\qquad\square$

# D  Additional Results

## D.1  Comparing Tails of Wider versus Deeper Models

To further demonstrate that Deep GP are heavy tailed and sharply peaked, Fig. 6 displays the *difference* between Deep GP marginal densities and the limiting GP marginal density, i.e.:

$$p_{\mathrm{DGP}}(y_1, y_2 \mid \mathbf{x}_1, \mathbf{x}_2) - p_{\mathrm{Lim.\ GP}}(y_1, y_2 \mid \mathbf{x}_1, \mathbf{x}_2).$$

Red areas correspond to values of $\mathbf{y}$ where the Deep GP has more density, while blue areas correspond to values where the limiting GP has more density. Note that Deep GP of all widths and depths have red values near the $[0, 0]$ mean (corresponding to a sharper peak than the limiting GP) and red values in the upper left and lower right quadrants (corresponding to heavier tails than the limiting GP).

We also note that deeper/narrower models have heavier tails and sharper peaks than shallower/wider models (Fig. 7). The left plot shows the difference between the marginal densities of Depth-3 and Depth-2 Deep GP (with the same first and second moments). The remaining plots show the difference between Depth-2 Deep GP of varying width (again, with the same first and second moments).

## D.2  Control Experiment: How Well Does NUTS Sample Deep GP Posteriors?

In order to determine if the NUTS sampler accurately captures Deep GP performance, we perform a control experiment on synthetic data. Specifically, we generate $N = 1000$ datasets from width $= 1$, width $= 2$, width $= 4$, and width $= 8$ Deep GP, sampling from these models at randomly-generated $\mathbf{x}$ values in a 4-dimensional input space. Each generating Deep GP has two layers: the first uses a RBF covariance, and the second uses the sum of 1-dimensional RBF covariances. We then train width $= 1$, width $= 2$, width $= 4$, and width $= 8$ Deep GP on each of the generated datasets, using half the data for training and half for testing. Our hypothesis is that width $= j$ models should at least achieve good test set performance on width $= j$ generated datasets.

Fig. 8 displays the test set log likelihood on the generated datasets. We see that width $\in \{2, 4, 8\}$ models tend to achieve similar performance on each of the datasets. On the other hand, the width $= 1$ models achieve significantly worse test set performance, even on the dataset generated by a width $= 1$ model. This suggests that the NUTS sampler is unable to converge to good posterior samples for width $= 1$ models, which may potentially explain the superior performance of width $= 2$ Deep GP observed in the Sec. 6.1 experiments.

## D.3  3-Layer Deep GP and 3-Layer Bayesian Neural Networks

We extend the experiments from Sec. 6.1 to 3-layer Deep GP and Bayesian neural networks. Specifically, we measure the test set log likelihood (Fig. 9 top) and training set kernel fit (Fig. 9 bottom) on 6 regression datasets. The Deep GP models use a standard RBF covariance function for the first layer and sums of 1-dimensional RBF covariances for the second and third layers. The neural networks add an additional $f_3(\cdot) = \frac{1}{\sqrt{H_2}}\mathbf{w}_3^\top\boldsymbol{\sigma}(\cdot) + \beta b_3$ layer on top of the Eq. (1) construction, where $H_2$ is the width of the second layer and the entries of $\mathbf{w}_3, b_3$ are i.i.d. unit Gaussian.

We compare models of different width, increasing the width of both hidden layers simultaneously. It is worth noting that changing the width of the first hidden layer affects the prior second moment of the Deep GP/neural network. Therefore, unlike the the experiments in Sec. 6.1, we are no longer ensuring that all models are moment-matched. Nevertheless, width has the same effect for 3-layer models as it does for 2-layer models. Width-2 Deep GP almost always outperform all other Deep GP, and neural networks achieve best log likelihood and kernel fit with $\leq 8$ hidden units per layer.

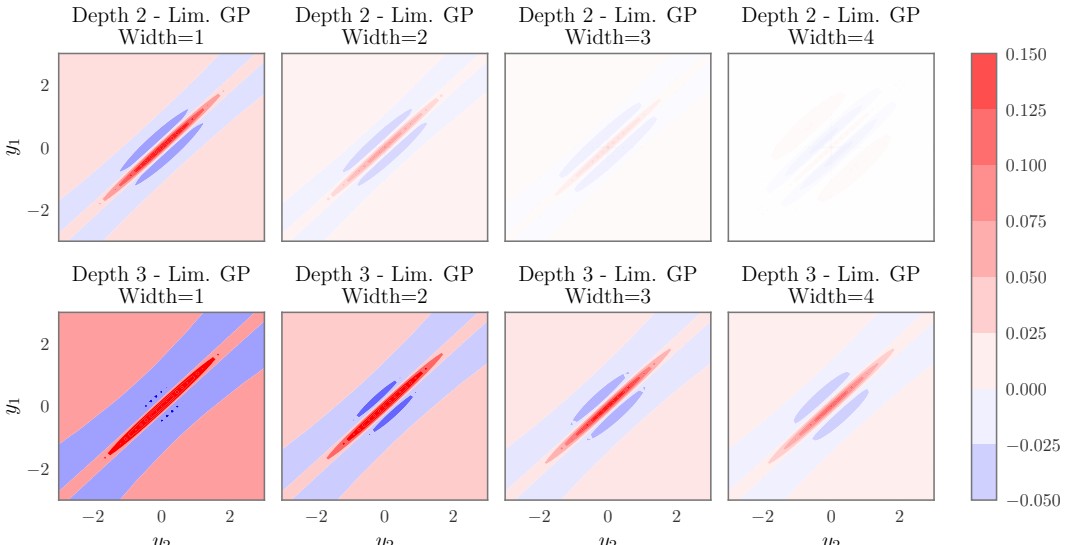

Figure 6: We depict the *difference* between the Deep GP marginal density $p_{\text{DGP}}(y_1, y_2 \mid \mathbf{x}_1, \mathbf{x}_2)$ and the limiting GP marginal density $p_{\text{Lim. GP}}(y_1, y_2 \mid \mathbf{x}_1, \mathbf{x}_2)$ on the $N = 2$ dataset $\mathbf{x}_1 = -0.5$, $\mathbf{x}_2 = 0.5$. Red regions correspond to values of $y_1, y_2$ where the Deep GP has more density, and vice versa for the blue regions. All Deep GP have heavier tails and a sharper peak than the limiting GP.

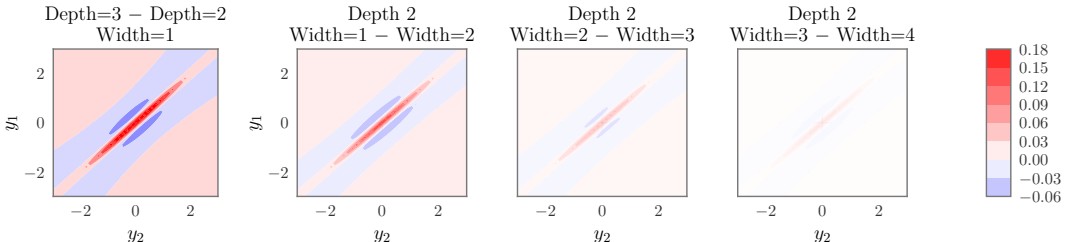

Figure 7: We depict the *difference* between the marginal densities of various Deep GP on the $N = 2$ dataset $\mathbf{x}_1 = -0.5$, $\mathbf{x}_2 = 0.5$. Red regions correspond to values of $y_1, y_2$ where the deeper/narrower Deep GP has more density, and vice versa for the blue regions. **Left:** Comparing Deep GP of different depth. The 3-layer (width-1) Deep GP has a sharper peak and heavier tails than the 2-layer model, as indicated by the red regions. **Right:** Comparing 2-layer Deep GP of different width. Width-$j$ models have sharper peaks and heavier tails than width-$j + 1$ models, as indicated by the red regions. All models have the same first and second moments.

Test Set Log Likelihood

| Generating Model | Width=1 | Width=2 | Width=4 | Width=8 |
|---|---|---|---|---|
| Width=1 | -0.836 | -0.760 | -0.765 | -0.767 |
| Width=2 | -1.040 | -0.794 | -0.798 | -0.812 |
| Width=4 | -0.930 | -0.778 | -0.773 | -0.776 |
| Width=8 | -0.889 | -0.818 | -0.816 | -0.816 |

Fitting Model

Figure 8: Control experiment to test how well NUTS sampling captures the Deep GP posterior. We generate datasets from width $\in \{1, 2, 4, 8\}$ Deep GP, and then fit width $\in \{1, 2, 4, 8\}$ Deep GP to these datasets. The width $\in \{2, 4, 8\}$ models achieve roughly the same test set log likelihood on each dataset (higher is better). Width $= 1$ Deep GP tend to achieve worse log likelihoods, even on a dataset generated from a width $= 1$ Deep GP.

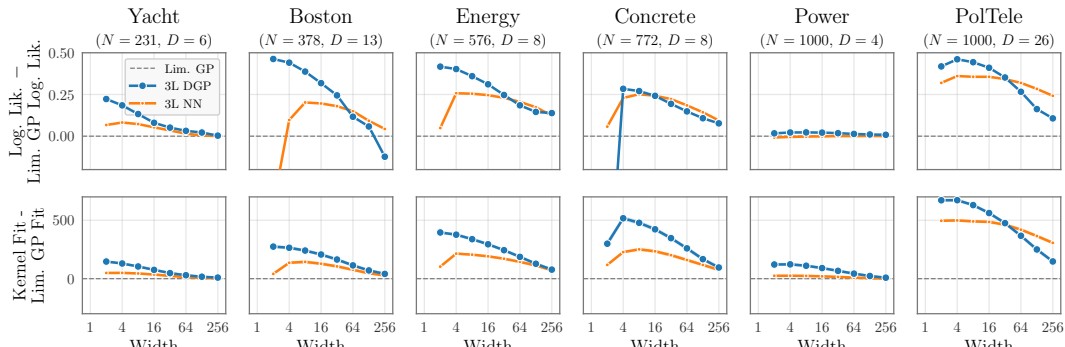

Figure 9: **Top:** Test set log likelihood (LL) of **3-layer Deep GP** (and neural networks) as a function of width on regression datasets (higher is better). Numbers are shifted so that 0 corresponds to the limiting GP log likelihood. **Bottom:** Fit of the posterior kernel $k(\mathbf{f}_2(\mathbf{f}_1(\cdot)), \mathbf{f}_2(\mathbf{f}_1(\cdot)))$ on the training data, as measured by Gaussian log marginal likelihood (higher is better). 0 corresponds to the limiting GP log marginal likelihood.

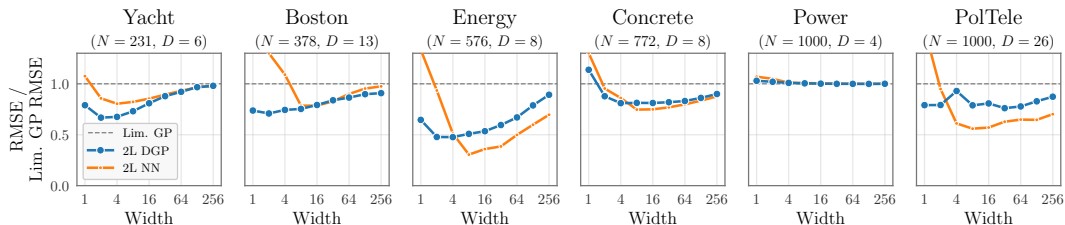

Figure 10: Test set root mean squared error (RMSE) of **2-layer** Deep GP (and neural networks) as a function of width on regression datasets (lower is better). Numbers are scaled so that 1 corresponds to the limiting GP RMSE.

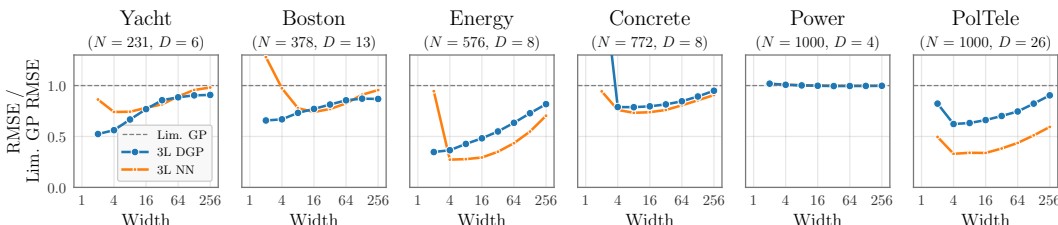

Figure 11: Test set root mean squared error (RMSE) of **3-layer** Deep GP (and neural networks) as a function of width on regression datasets (lower is better). Numbers are scaled so that 1 corresponds to the limiting GP RMSE.

Table 2: Root mean squared error (RMSE) of Deep GP on UCI regression datasets as a function of depth (lower is better). All models for a given dataset have the same prior covariance.

| Depth | Yacht $(N = 231, D = 6)$ | Boston $(N = 378, D = 13)$ | Energy $(N = 576, D = 8)$ | Concrete $(N = 772, D = 8)$ | Power $(N = 1000, D = 4)$ | PolTele $(N = 1000, D = 26)$ |
|---|---|---|---|---|---|---|
| 1 | 0.327 | 0.643 | 0.267 | 0.424 | 0.241 | 0.305 |
| 2 | 0.240 | 0.480 | 0.183 | **0.350** | **0.235** | 0.229 |
| 3 | **0.229** | **0.426** | **0.169** | 0.438 | 0.236 | **0.218** |

## D.4 Additional Figures and Tables

Figs. 10 and 11 report the test set root mean squared error (RMSE) of 2-layer and 3-layer models as a function of width (lower is better). As with log likelihood, we find that width generally harms RMSE. Table 2 reports the test set RMSE for Deep GP of various depth (controlling the first and second prior moments, as in Table 1). Again, depth is generally beneficial with regards to RMSE.

# E    Proof of Theorem 1

Here we prove our main theoretical result (Thm. 1), which states that Deep GP converge to (single-layer) GP in the infinite width limit. Following Matthews et al. [64], we will prove that random processes with countable index sets converge in distribution to GP with countable index sets. Thus, it is sufficient to prove that the marginals of the random process converge in distribution to multivariate Gaussians [e.g. 14]. While a more general treatment may be of theoretical interest, the limitation to countable index sets is sufficient for machine learning applications where we are often concerned with a finite set of events.

Before arriving at a general result, we begin with specialized proofs for two subclasses of Deep GP: 1) those that use *additively-decomposing prior covariance functions* in each GP layer (Observation 1), and 2) those that use *isotropic prior covariance functions* (Observation 2). We note that these two cases include many "textbook" covariance functions (i.e. the kernels described by Genton [40] or Rasmussen and Williams [77, Ch. 4]). Afterwards, we present a more general result. We note that the assumptions required for the general result are rather minimal, and are indeed satisfied by most Deep GP architectures (or arbitrarily-precise approximations thereof).

## E.1    Warmup 1: Deep GP with Additive and/or Isotropic Covariance Functions

The first case we will explore is Deep GP with covariance functions that decompose additively. We will assume that the output of the additive composition is scaled to account for the input dimensionality. This additive decomposition suggests a straightforward application of the strong law of large numbers.

**Observation 1.** *Let $f_2(\mathbf{f}_1(\mathbf{x}))$ be a 2-layer zero-mean Deep GP, where $k_2(\cdot, \cdot) : \mathbb{R}^{H_1} \times \mathbb{R}^{H_1} \to \mathbb{R}$ can be written in the form:*

$$k_2(\mathbf{z}, \mathbf{z}') = \frac{1}{H_1} \sum_{i=1}^{H_1} k_2^{(comp)}(z_i, z_i'),\tag{18}$$

*where $k_2^{(comp)}(\cdot, \cdot) : \mathbb{R} \times \mathbb{R} \to \mathbb{R}$ is a positive definite function that is bounded by some polynomial:*

$$|k_2^{(comp)}(z, z')| \leq \sum_{j=0}^{R} \sum_{k=0}^{j} a_{j,k} |z^{j-k} z'^k|$$

*for some $R < \infty$ and constants $a_{j,k} > 0$. Additionally, assume that $|k_1(\mathbf{x}, \mathbf{x}')| < \infty$ for all finite $\mathbf{x}$, $\mathbf{x}'$. Then the conditional covariance $\mathbb{E}[f_2(\mathbf{f}_1(\mathbf{x})) f_2(\mathbf{f}_1(\mathbf{x}')) \mid \mathbf{f}_1(\mathbf{x}), \mathbf{f}_1(\mathbf{x}')]$ becomes almost surely constant as $H_1 \to \infty$ for finite $\mathbf{x}, \mathbf{x}'$.*

*Proof.* We have that

$$\lim_{H_1 \to \infty} \mathbb{E}\left[ f_2(\mathbf{f}_1(\mathbf{x})) f_2(\mathbf{f}_1(\mathbf{x}')) \mid \mathbf{f}_1(\mathbf{x}) \mathbf{f}_1(\mathbf{x}') \right]$$

$$= \lim_{H_1 \to \infty} k_2(\mathbf{f}_1(\mathbf{x}), \mathbf{f}_1(\mathbf{x}')) = \lim_{H_1 \to \infty} \frac{1}{H_1} \sum_{i=1}^{H_1} k_2^{(comp)}(f_1^{(i)}(\mathbf{x}), f_1^{(i)}(\mathbf{x}')).\tag{19}$$

Note that all the $k_2^{(comp)}(f^{(i)}(\mathbf{x}), f^{(i)}(\mathbf{x}'))$ terms are i.i.d. by construction. Moreover, since $k_1(\mathbf{x}, \mathbf{x}')$ is finite for all finite $\mathbf{x}, \mathbf{x}'$, all moments of $f_1^{(i)}(\mathbf{x}), f_1^{(i)}(\mathbf{x}')$ will also be finite. Using our assumptions

on $k_2^{(\text{comp})}(\cdot, \cdot)$, we have:

$$\left| \mathop{\mathbb{E}}_{f_1^{(i)}(\cdot)} \left[ k_2^{(\text{comp})}(f_1^{(i)}(\mathbf{x}), f_1^{(i)}(\mathbf{x}')) \right] \right| \leq \mathop{\mathbb{E}}_{f_1^{(i)}(\cdot)} \left[ \left| k_2^{(\text{comp})}(f_1^{(i)}(\mathbf{x}), f_1^{(i)}(\mathbf{x}')) \right| \right]$$

$$\leq \sum_{j=0}^{R} \sum_{k=0}^{j} a_{j,k} \mathop{\mathbb{E}}_{f_1^{(i)}(\cdot)} \left[ \left| f_1^{(i)}(\mathbf{x})^{j-k} f_1^{(i)}(\mathbf{x}')^k \right| \right]$$

$$\leq \sum_{j=0}^{R} \sum_{k=0}^{j} a_{j,k} \sqrt{ \mathop{\mathbb{E}}_{f_1^{(i)}(\cdot)} \left[ \left| f_1^{(i)}(\mathbf{x})^{2j-2k} f_1^{(i)}(\mathbf{x}')^{2k} \right| \right] }$$

$$< \infty. \qquad \text{(moments of } f_1^{(i)}(\mathbf{x}), f_1^{(i)}(\mathbf{x}') \text{ are finite)}$$

Therefore, we can apply the strong law of large numbers to the limit in Eq. (19) which thus is almost surely constant. $\qquad \square$

Neural networks are one common case of a Deep GP with additive covariance functions. Combining Observation 1 with Lemma 1 gives the classic GP convergence result first discovered by Neal [69]. We note however that, when additive structure exists, we do not need to rely on the covariance perspective of Lemmas 1 and 2, as we can prove GP convergence using the central limit analysis of [56, 64, 69].

### E.2  Warmup 2: Deep GP with Continuous Isotropic Covariance Functions

The most common Deep GP architectures use RBF or Matérn covariance functions for the GP layers [e.g. 19, 26, 27, 46, 79]. These covariance functions belong to the class of *isotropic kernels*, which are covariances that can be written as a function of Euclidean distance:

$$k(\mathbf{z}, \mathbf{z}') = \varphi(\|\mathbf{z} - \mathbf{z}'\|_2^2).$$

Similar to Observation 1, our analysis of isotropic Deep GP relies on the strong law of large numbers. Again, we will assume that the covariance functions are scaled to account for the input dimensionality.

**Observation 2.** *Define* $\mathbb{A} \triangleq \cup_{d=1}^{\infty}(\mathbb{R}^d \times \mathbb{R}^d)$. *Let* $f_2(\mathbf{f}_1(\mathbf{x}))$ *be a 2-layer zero-mean Deep GP, where* $k_2(\cdot, \cdot) : \mathbb{A} \to \mathbb{R}$ *is a continuous isotropic kernel that can be written in the form:*

$$k_2(\mathbf{z}, \mathbf{z}') = \begin{cases} \varphi\left((\mathbf{z} - \mathbf{z}')^2\right) & \mathbf{z}, \mathbf{z}' \in \mathbb{R} \\ \varphi\left(\frac{1}{2}\|\mathbf{z} - \mathbf{z}'\|_2^2\right) & \mathbf{z}, \mathbf{z}' \in \mathbb{R}^2 \\ \varphi\left(\frac{1}{3}\|\mathbf{z} - \mathbf{z}'\|_2^2\right) & \mathbf{z}, \mathbf{z}' \in \mathbb{R}^3 \\ \varphi\left(\frac{1}{4}\|\mathbf{z} - \mathbf{z}'\|_2^2\right) & \mathbf{z}, \mathbf{z}' \in \mathbb{R}^4 \\ \vdots & \end{cases} . \tag{20}$$

*Additionally, assume that* $|k_1(\mathbf{x}, \mathbf{x}')| < \infty$ *for all finite* $\mathbf{x}, \mathbf{x}'$. *Then the conditional covariance* $\mathbb{E}[f_2(\mathbf{f}_1(\mathbf{x}))f_2(\mathbf{f}_1(\mathbf{x}')) \mid \mathbf{f}_1(\mathbf{x}), \mathbf{f}_1(\mathbf{x}')]$ *becomes almost surely constant as* $H_1 \to \infty$ *for finite* $\mathbf{x}, \mathbf{x}'$.

*Proof.* We have that

$$\lim_{H_1 \to \infty} \mathbb{E}\left[ f_2(\mathbf{f}_1(\mathbf{x}))f_2(\mathbf{f}_1(\mathbf{x}')) \mid \mathbf{f}_1(\mathbf{x})\mathbf{f}_1(\mathbf{x}') \right]$$

$$= \lim_{H_1 \to \infty} k_2(\mathbf{f}_1(\mathbf{x}), \mathbf{f}_1(\mathbf{x}')) = \lim_{H_1 \to \infty} \varphi\left( \frac{1}{H_1} \|\mathbf{f}_1(\mathbf{x}) - \mathbf{f}_1(\mathbf{x}')\|_2^2 \right)$$

$$= \varphi\left( \lim_{H_1 \to \infty} \frac{1}{H_1} \sum_{i=1}^{H_1} \left( f_1^{(i)}(\mathbf{x}) - f_1^{(i)}(\mathbf{x}') \right)^2 \right) \tag{21}$$

where we can move the limit inside $\varphi(\cdot)$ by the continuity assumption. Note that $f_1^{(i)}(\mathbf{x}) - f_1^{(i)}(\mathbf{x}') \overset{\text{i.i.d}}{\sim} \mathcal{N}(0, \sigma^2)$, where $\sigma^2 = k_1(\mathbf{x}, \mathbf{x}) + k_1(\mathbf{x}', \mathbf{x}') - 2k_1(\mathbf{x}, \mathbf{x}')$. Since $k_1(\mathbf{x}, \mathbf{x}')$ is finite—and thus $\sigma^2 = \mathbb{E}[(f_1^{(i)}(\mathbf{x}) - f_1^{(i)}(\mathbf{x}'))^2]$ is finite—Eq. (21) is almost surely constant by the strong law of large numbers. $\qquad \square$

Combining Observation 2 with Lemma 2, we have that Deep GP with isotropic covariance functions become GP in their infinite width limit.

### E.3   Assumptions Required for a General Result

We note that the most common Deep GP/neural network architectures either decompose additively or use isotropic covariance functions [1, 2, 19, 27, 79]. In these cases, Observations 1 and 2 are sufficient to prove GP convergence. To prove a more general result, we will need to make an additional set of assumptions about the GP covariance functions. We emphasize that these assumptions are sufficient but not necessary conditions, yet they hold for most Deep GP architectures (or arbitrarily-accurate approximations thereof). Informally, these assumptions are:

1) each covariance function *"scales reasonably"* with the dimensionality of its inputs;

2) each covariance function has a *Lesbegue-Stieltjes representation*; and

3) each covariance function is *bounded.*

These assumptions are—in practice—very minimal. The first item ensures that the prior covariance doesn't "blow up" for high-dimensional data (e.g. the covariance does not converge to a constant for all inputs). The second item admits all but the most pathological covariance functions [87]. The last item may at first seem unreasonable, since some common covariance functions are unbounded (e.g. linear kernel, polynomial kernel, etc.). However, we note that many covariance functions are bounded on any compact domain, and therefore from a practical perspective we can approximate any of these unbounded covariances to any arbitrary precision.

**Assumption 1** (Covariance functions have Lesbegue-Stieltjes representations with compact spectral support)**.** We assume that all covariance functions of interest $k(\mathbf{z}, \mathbf{z}') : \mathbb{R}^D \times \mathbb{R}^D \to \mathbb{R}$ can be represented by a Lesbegue-Stieltjes integral of the following form:

$$k(\mathbf{z}, \mathbf{z}') = \int \rho \left( \frac{1}{D} \sum_{i=1}^{D} \phi \left( z_i \xi_i - z_i' \xi_i' \right) \right) \, \mathrm{d}\mu((\xi_1, \xi_1'), \ldots, (\xi_D, \xi_D')), \tag{22}$$

where

- $\mu((\xi_1, \xi_1'), \ldots, (\xi_D, \xi_D'))$ is a positive definite function of bounded variation and compact support: i.e.: there exists some constant $C < \infty$ such that $\mu((\mathbb{R} \times \mathbb{R})^D - ([-C, C] \times [-C, C])^D) = 0$;

- $\phi : \mathbb{R} \to \mathbb{R}$ is bounded above and below by a finite polynomial; and

- $\rho(z) : \mathbb{R} \to \mathbb{R}$ is a bounded and continuous function.

We will show that this formulation—though complex—admits additive kernels, isotropic kernels, and most other non-pathological kernels (or arbitrarily-precise approximations thereof).

**Assumption 2** (Covariance functions scale with dimensionality)**.** Let $\mathbb{A} = \cup_{d=1}^{\infty} (\mathbb{R}^d \times \mathbb{R}^d)$. We assume that the covariance function $k(\cdot, \cdot) : \mathbb{A} \to \mathbb{R}$ scales *reasonably with dimensionality*; that is, it can be written in the following form:

$$k(\mathbf{z}, \mathbf{z}') = \begin{cases} k^{(1)}(\mathbf{z}, \mathbf{z}'), & \mathbf{z}, \mathbf{z}' \in \mathbb{R}, \\ k^{(2)}(\mathbf{z}, \mathbf{z}'), & \mathbf{z}, \mathbf{z}' \in \mathbb{R}^2, \\ k^{(3)}(\mathbf{z}, \mathbf{z}'), & \mathbf{z}, \mathbf{z}' \in \mathbb{R}^3, \\ \vdots \end{cases}$$

where the $k^{(j)}(\mathbf{z}, \mathbf{z}')$ satisfy Assumption 1:

$$k^{(j)}(\mathbf{z}, \mathbf{z}') = \int \rho \left( \frac{1}{j} \sum_{i=1}^{j} \phi \left( z_i \xi_i - z_i' \xi_i' \right) \right) \, \mathrm{d}\mu_j((\xi_1, \xi_1'), \ldots, (\xi_j, \xi_j')),$$

and the measures $\mu_j$ satisfy the Kolmogorov consistency condition:

$$\mu_{j+k} \left( \mathcal{E} \times \mathbb{R}^k \right) = \mu_j \left( \mathcal{E} \right) \text{ for every } j, k \geq 1, \text{ and every Borel set } \mathcal{E} \subset \mathbb{R}^j.$$

### E.4 An Exhaustive Discussion About Assumptions 1 and 2

We again reiterate that Assumptions 1 and 2 are not necessary conditions. If more is known about the Deep GP architecture (e.g. all covariance functions are additive or isotropic, etc.), then it is possible to use a smaller and simpler set of assumptions. We nevertheless argue that—in practice, Assumptions 1 and 2 are indeed very general.

**All but the most pathological covariance functions can be expressed (or well-approximated) by Assumption 1.** If we choose $\rho(\cdot) = \cos(\cdot)$, $\phi(\cdot)$ to be the identity function, and $\mu$ to be any measure of bounded variation, then Eq. (22) reduces to:

$$k(\mathbf{z}, \mathbf{z}') = \int \cos\left(\frac{1}{D}\sum_{i=1}^{D}\left(\mathbf{z}^\top \boldsymbol{\xi} - \mathbf{z}'^\top \boldsymbol{\xi}'\right)\right) \mathrm{d}\mu(\boldsymbol{\xi}, \boldsymbol{\xi}'), \tag{23}$$

where $\boldsymbol{\xi}$ and $\boldsymbol{\xi}'$ are equal to $[\xi_1, \ldots, \xi_D]$ and $[\xi_1', \ldots, \xi_D']$, respectively. This is a Fourier-Stieljes integral, and almost all (bounded) covariance functions encountered in the machine learning literature can be written in this form [40]. Such covariance functions are known as *harmonizable kernels*. When the measure $\mu$ is concentrated on the diagonal, then Eq. (23) reduces to Bochner's representation of stationary covariance functions [77, Ch. 4]. More generally, a non-diagonal $\mu$ results in non-stationary covariance functions, and Yaglom [87, Sec. 26.4] argues that (bounded) covariances that *cannot* be expressed by Eq. (23) tend to be pathological in nature.

Of course, any covariance function that can be expressed as Eq. (23) is necessarily *bounded*, since $\cos(\cdot)$ is bounded and the measure $\mu$ has bounded variation by assumption. The most common unbounded covariance functions are dot-product kernels, such as the linear kernel ($\beta^2 + \mathbf{x}^\top \mathbf{x}'$ for some constant $\beta > 0$) or the "ReLU kernel" ($\beta^2 + \boldsymbol{\sigma}(\mathbf{x})^\top \boldsymbol{\sigma}(\mathbf{x}')$, where $\boldsymbol{\sigma}(\cdot) = \max\{\mathbf{0}, \cdot\}$). Importantly, these covariance functions meet the additive structure condition of Observation 1, and so they do not require the general treatment of Assumptions 1 and 2. However, we would also note that these covariances are bounded on any compact domain, so we can approximate them to arbitrary precision by replacing $\mathbf{x}$ with $(\mathbf{x}/\|\mathbf{x}\|_\infty)\min\{\|\mathbf{x}\|_\infty, B\}$ for any $B < \infty$ and similarly for $\mathbf{x}'$.

The other simplifying assumption is that the spectral measure $\mu$ has compact support. Again, even if $k(\cdot, \cdot)$ corresponds to a spectral measure with infinite support, it can be approximated to arbitrary precision by replacing $\boldsymbol{\xi}$ with $(\boldsymbol{\xi}/\|\boldsymbol{\xi}\|_\infty)\min\{\|\boldsymbol{\xi}\|_\infty, B\}$ for some constant $B$.

**Assumption 2 captures natural ways of scaling to dimensionality.** The two requirements of Assumption 2 are mechanisms for defining reasonable sequences of covariance functions. The consistency requirement on $\mu_j$ ensures that covariance functions do not "change significantly" as dimensionality increases. The $1/j$ term simply prevents covariances from becoming unbounded or degenerate as $j \to \infty$. We note that—by choosing appropriate $\rho(\cdot)$ and $\phi(\cdot)$ functions in Eq. (22)— this $1/j$ term can correspond to "natural" scaling rates. For example:

- If $k(\cdot, \cdot)$ is isotropic, then Eq. (22) can be reduced to:

$$k_j(\mathbf{z}, \mathbf{z}') = \varphi\left(\frac{1}{j}\sum_{i=1}^{j}(z_i - z_i')^2\right) = \varphi\left(\frac{1}{j}\|\mathbf{z}_i - \mathbf{z}_i'\|^2\right),$$

  where $\varphi(\cdot)$ is a continuous positive definite function. We get this by setting $\rho(\cdot) = \varphi(\cdot)$, $\phi(\cdot) = (\cdot)^2$, and by setting $\mu_j$ to be atomic. This is the scaling of isotropic covariances studied in Appx. E.2.

- If $k(\cdot, \cdot)$ is additive, then Eq. (22) can take the form:

$$k_j(\mathbf{z}, \mathbf{z}') = \frac{1}{j}\sum_{i=1}^{j}\int \cos\left(z_i\xi_i - z_i'\xi_i'\right)\mathrm{d}\mu(\xi_i, \xi_i'), \tag{24}$$

  where $\mu$ is of bounded variation. We get this by setting $\rho(z) = (z/|z|)\min\{|z|, B\}$ for some sufficiently large constant $B$ and by setting $\phi(\cdot) = \cos(\cdot)$.[5] This is now the sum of 1D harmonizable covariance functions using the scaling studied in Appx. E.1.

---

[5]Since the integral $\int \cos\left(z_i\xi_i - z_i'\xi_i'\right)\mathrm{d}\mu(\xi_i, \xi_i')$ is bounded, there exists some constant $B$ such that $\rho(z) = z/|z|\min\{|z|, B\}$ is effectively equal to the identity function in Eq. (24).

## E.5 A General Result

With Assumptions 1 and 2, we show that the conditional covariance of a 2-layer zero-mean Deep GP becomes almost surely constant in the limit of infinite-width.

**Lemma 3.** *Let $f_2(\mathbf{f}_1(\mathbf{x}))$ be a 2-layer zero-mean Deep GP, where $k_2(\cdot, \cdot)$ satisfies Assumptions 1 and 2. The conditional covariance $\mathbb{E}[f_2(\mathbf{f}_1(\mathbf{x}))f_2(\mathbf{f}_1(\mathbf{x}')) \mid \mathbf{f}_1(\mathbf{x}), \mathbf{f}_1(\mathbf{x}')]$ becomes almost surely constant as $H_1 \to \infty$.*

As with our warmups (Observations 1 and 2), the proof of Lemma 3 essentially boils down to applying the strong law of large numbers. The primary complication of this proof is ensuring that Assumptions 1 and 2 satisfy the conditions necessary to invoke the strong law.

*Proof.* By Assumptions 1 and 2, the limiting conditional covariance can be written as:

$$\lim_{H_1 \to \infty} \mathbb{E}\left[f_2(\mathbf{f}_1(\mathbf{x}))f_2(\mathbf{f}_1(\mathbf{x}')) \mid \mathbf{f}_1(\mathbf{x}), \mathbf{f}_1(\mathbf{x}')\right]$$

$$= \lim_{H_1 \to \infty} k_2(\mathbf{f}_1(\mathbf{x}), \mathbf{f}_1(\mathbf{x}'))$$

$$= \lim_{H_1 \to \infty} \int \rho\left(\frac{1}{H_1} \sum_{i=1}^{H_1} \phi\left(f_1^{(i)}(\mathbf{x})\xi_i - f_1^{(i)}(\mathbf{x}')\xi_i'\right)\right) \mathrm{d}\mu_{(H_1)}((\xi_1, \xi_1'), \ldots, (\xi_{H_1}, \xi_{H_1}')). \quad (25)$$

By the consistency requirement in Assumption 2, we know that there exists a unique probability measure $\mu_\infty$ on the Borel product sigma-algebra over $\mathbb{R}^\infty$ such that, for any $H_1 \in \mathbb{N}$ and any Borel subset $\mathcal{E} \in \mathbb{R}^{H_1}$, we have $\mu_{H_1}(\mathcal{E}) = \mu_\infty(\mathcal{E} \times \mathbb{R} \times \mathbb{R} \times \ldots)$. Therefore, we can rewrite Eq. (25) as

$$\lim_{H_1 \to \infty} \int g_{\mathbf{f}_1(\mathbf{x}), \mathbf{f}_1(\mathbf{x}')}^{(H_1)}(\boldsymbol{\xi}, \boldsymbol{\xi}') \, \mathrm{d}\mu_\infty(\boldsymbol{\xi}, \boldsymbol{\xi}'), \quad (26)$$

where $\mathbf{f}_1(\mathbf{x})$, $\mathbf{f}_1(\mathbf{x}')$, $\boldsymbol{\xi}$, and $\boldsymbol{\xi}'$ are infinite dimensional vectors ($\mathbf{f}_1(\mathbf{x}) = [f_1^{(1)}(\mathbf{x}), f_1^{(2)}(\mathbf{x}), \ldots]$, $\boldsymbol{\xi} = [\xi_1, \xi_2, \ldots]$), and $g_{\mathbf{f}_1(\mathbf{x}), \mathbf{f}_1(\mathbf{x}')}^{(H_1)}$ is the random function given by

$$g_{\mathbf{f}_1(\mathbf{x}), \mathbf{f}_1(\mathbf{x}')}^{(H_1)}(\boldsymbol{\xi}, \boldsymbol{\xi}') = \rho\left(\frac{1}{H_1} \sum_{i=1}^{H_1} \phi\left(f_1^{(i)}(\mathbf{x})\xi_i - f^{(i)}(\mathbf{x}')\xi_i'\right)\right). \quad (27)$$

Note that $g_{\mathbf{f}_1(\mathbf{x}), \mathbf{f}_1(\mathbf{x}')}^{(H_1)}(\boldsymbol{\xi}, \boldsymbol{\xi}')$ is bounded (because $\rho(\cdot)$ is bounded). Moreover, since each of the $\mu_j$ have compact support, $\mu_\infty$ will have compact support as well. Therefore, we can consider the domain of $g_{\mathbf{f}_1(\mathbf{x}), \mathbf{f}_1(\mathbf{x}')}^{(H_1)}(\boldsymbol{\xi}, \boldsymbol{\xi}')$ to be $[-C, C]^D \times [-C, C]^D$ for some constant $C < \infty$, and the range to be $[-B, B]$ for some constant $B < \infty$. Now consider the limit of Eq. (27):

$$\lim_{H_1 \to \infty} g_{\mathbf{f}_1(\mathbf{x}), \mathbf{f}_1(\mathbf{x}')}^{(H_1)}(\boldsymbol{\xi}, \boldsymbol{\xi}') = \rho\left(\lim_{H_1 \to \infty} \frac{1}{H_1} \sum_{i=1}^{H_1} \phi\left(f_1^{(i)}(\mathbf{x})\xi_i - f_1^{(i)}(\mathbf{x}')\xi_i'\right)\right), \quad (28)$$

where we can bring the limit inside $\rho$ by continuity. Consider fixed inputs $\boldsymbol{\xi}, \boldsymbol{\xi}' \in [-C, C]^\infty$. The $f^{(i)}(\cdot)$ terms are i.i.d. zero-mean Gaussian by construction, and have finite variance because $k_1(\cdot, \cdot)$ is finite almost everywhere (Assumption 1). Additionally, the $\xi_i$ and $\xi_i'$ terms lie in the finite interval $[-C, C]$. Consequentially, $f_1^{(i)}(\mathbf{x})\xi_i - f_1^{(i)}(\mathbf{x}')\xi_i'$ are Gaussian random variables with zero mean and bounded variance. Moreover, $\phi(\cdot)$ is bounded above and below by a finite polynomial (Assumption 1), and thus $\mathrm{Var}[\phi(f_1^{(i)}(\mathbf{x})\xi_i - f_1^{(i)}(\mathbf{x}')\xi_i')]$ is bounded by a finite linear combination of the moments of $f_1^{(i)}(\mathbf{x})\xi_i - f_1^{(i)}(\mathbf{x}')\xi_i'$. Since the moments of a Gaussian are positive polynomial functions of the variance, we have that $\mathrm{Var}[\phi(f_1^{(i)}(\mathbf{x})\xi_i - f_1^{(i)}(\mathbf{x}')\xi_i')]$ is bounded, and thus

$$\left|\mathbb{E}\left[\phi\left(f_1^{(i)}(\mathbf{x})\xi_i - f_1^{(i)}(\mathbf{x}')\xi_i'\right)\right]\right| < \infty, \qquad \sum_{i=1}^{\infty} \frac{1}{i^2} \mathrm{Var}\left[\phi\left(f_1^{(i)}(\mathbf{x})\xi_i - f_1^{(i)}(\mathbf{x}')\xi_i'\right)\right] < \infty.$$

We therefore satisfy the strong law of large numbers conditions, and so Eq. (28) converges to a constant almost surely for any $\boldsymbol{\xi}, \boldsymbol{\xi}' \in [-C, C]^\infty$. In other words,

$$\lim_{H_1 \to \infty} g_{\mathbf{f}_1(\mathbf{x}), \mathbf{f}_1(\mathbf{x}')}^{(H_1)}(\boldsymbol{\xi}, \boldsymbol{\xi}') = \lim_{H_1 \to \infty} \frac{1}{H_1} \sum_{i=1}^{H_1} \phi\left(f_1^{(i)}(\mathbf{x})\xi_i - f_i^{(i)}(\mathbf{x}')\xi_i'\right) = \text{const. a.s.} \quad (29)$$

Since this holds for all $\boldsymbol{\xi}, \boldsymbol{\xi}' \in [-C, C]^\infty$, we have:

$$\int \lim_{H_1 \to \infty} g^{(H_1)}_{\mathbf{f}_1(\mathbf{x}), \mathbf{f}_1(\mathbf{x}')} (\boldsymbol{\xi}, \boldsymbol{\xi}') \, \mathrm{d}\mu_\infty(\boldsymbol{\xi}, \boldsymbol{\xi}') = \text{const. a.s.} \tag{30}$$

To finish off, we must show that dominated convergence implies that

$$\lim_{H_1 \to \infty} \int g^{(H_1)}_{\mathbf{f}_1(\mathbf{x}), \mathbf{f}_1(\mathbf{x}')} (\boldsymbol{\xi}, \boldsymbol{\xi}') \, \mathrm{d}\mu_\infty(\boldsymbol{\xi}, \boldsymbol{\xi}') = \int \lim_{H_1 \to \infty} g^{(H_1)}_{\mathbf{f}_1(\mathbf{x}), \mathbf{f}_1(\mathbf{x}')} (\boldsymbol{\xi}, \boldsymbol{\xi}') \, \mathrm{d}\mu_\infty(\boldsymbol{\xi}, \boldsymbol{\xi}'). \tag{31}$$

For all $H_1 \in \mathbb{N}$, we have that $\left| g^{(H_1)}_{\mathbf{f}_1(\mathbf{x}), \mathbf{f}_1(\mathbf{x}')} \right| \leq B$ is bounded and therefore trivially dominated by $\int B \, \mathrm{d}\mu_\infty(\boldsymbol{\xi}, \boldsymbol{\xi}') < \infty$. Now consider any fixed value of $\mathbf{f}_1(\mathbf{x}), \mathbf{f}_1(\mathbf{x}')$. By Eq. (29), we have that $g^{(H_1)}_{\mathbf{f}_1(\mathbf{x}), \mathbf{f}_1(\mathbf{x}')} (\boldsymbol{\xi}, \boldsymbol{\xi}')$ will converge pointwise with respect to $\boldsymbol{\xi}, \boldsymbol{\xi}'$ except when $\mathbf{f}_1(\mathbf{x}), \mathbf{f}_1(\mathbf{x}')$ comes from a set of measure $0$. Therefore, Eq. (31) holds with probability $1$ (where the probablity is taken with respect to $\mathbf{f}_1(\mathbf{x}), \mathbf{f}_1(\mathbf{x}')$). Combining Eqs. (26), (30) and (31) completes the proof. $\square$

The proof of Thm. 1 follows from applying Lemmas 2 and 3.

**Theorem 1 (Restated).** *Let $f_L \circ \ldots \circ f_1 (\mathbf{x})$ be a zero-mean Deep GP (Eq. 2), where each layer satisfies Assumptions 1 and 2. Then $\lim_{H_{L-1} \to \infty} \cdots \lim_{H_1 \to \infty} f_L \circ \ldots \circ \mathbf{f}_1 (\mathbf{x})$ converges in distribution to a (single-layer) GP.*

*Proof.* In the two layer case, combining Lemmas 2 and 3 gives us:

$$\lim_{H_1 \to \infty} \mathbb{E} \left[ \exp \left( i \mathbf{t}^\top \mathbf{f}_2 \right) \right] = \exp \left( -\frac{1}{2} \mathbf{t}^\top \mathbb{E} \left[ \mathbf{f}_2 \mathbf{f}_2^\top \right] \mathbf{t} \right) \quad \text{for all } \mathbf{t} \in \mathbb{R}^N. \tag{32}$$

Note that this is the characteristic function of a zero-mean multivariate Gaussian with covariance $\mathbb{E}[\mathbf{f}_2 \mathbf{f}_2^\top]$. Thus by Lévy's continuity theorem, $\mathbf{f}_2$ converges in distribution to $\mathcal{N}(\mathbf{0}, \mathbb{E}[\mathbf{f}_2 \mathbf{f}_2^\top])$. Since this is true for any finite marginal $\mathbf{f}_2$, the Deep GP $f_2(\mathbf{f}_1(\cdot))$ converges in distribution to a (single-layer) Gaussian process. A simple induction extends this to multiple layers. $\square$

### E.6 Comparison to Agrawal et al. [1].

Agrawal et al. [1, Thm. 8] also study infinite-width limits of Deep GP, though their analysis is restricted to a sub-class of models. Specifically, they focus on infinitely-wide neural networks with finite bottleneck layers—a specific class of Deep GP that they refer to as *bottleneck NNGP*. As the width of the bottleneck layers grow, these models become neural networks with infinite width in all layers. Coupling this with the analysis of infinitely-wide neural networks [56, 64], we have that bottleneck NNGP converge to standard GP in the limit of infinite width. However, the authors note that not every Deep GP can be expressed by the bottleneck NNGP architecture [1, Remark 7], and so their analysis is not sufficient to prove that all Deep GP converge to GP.

It is worth considering whether this strategy can be applied to other architectures—i.e. what Deep GP can be reduced to infinite-width neural networks with bottlenecks. For example, Cutajar et al. [24] study Deep GP with isotropic covariances. They convert each GP layer into neural network-like layers using random Fourier features [76]. However, their model is not exactly equivalent to a neural network, and so the analysis of Agrawal et al. [1] does not immediately apply. Moreover, it is not obvious how to express nonstationary GP as neural network-like architectures with modular width.

Finally, we remark that the strategy of Agrawal et al. [1, Thm. 8] is in some sense the opposite of what is explored in this paper. They and others [24, 31, 33] reduce certain classes of Deep GP to infinitely-wide neural networks with bottlenecks; conversely, we reduce neural networks to Deep GP.

## F Proofs of Theorems 2 and 3

Both Thms. 2 and 3 use the same two-step strategy presented for Lemma 1. We will decompose an expectation using the law of total expectation, and then apply Jensen's inequality.

**Theorem 2 (Restated).** *Let $f_L \circ \ldots \circ \mathbf{f}_1(\cdot)$ be a zero-mean Deep GP. Given a finite set of inputs $\mathbf{X} = [\mathbf{x}_1, \ldots, \mathbf{x}_N]$, define $\mathbf{f}_\ell = [(f_\ell \circ \ldots \circ \mathbf{f}_1(\mathbf{x}_1)), \ldots, (f_\ell \circ \ldots \circ \mathbf{f}_1(\mathbf{x}_N))]$ for $\ell \in [1, L]$, and define $\mathbf{K}_{lim} = \mathbb{E}_{\mathbf{f}_L}[\mathbf{f}_L \mathbf{f}_L^\top]$. Then, $p(\mathbf{f}_L = \mathbf{0}) > \mathcal{N}(\mathbf{g} = \mathbf{0}; \mathbf{0}, \mathbf{K}_{lim})$.*

*Proof.* We first produce a bound on the characteristic function of $\mathbf{f}_L$:

$$\mathbb{E}_{\mathbf{f}_L}\left[\exp(i\mathbf{t}^\top \mathbf{f}_L)\right] = \mathbb{E}_{\mathbf{F}_1}\left[\mathbb{E}_{\mathbf{F}_2|\mathbf{F}_1}\left[\ldots \mathbb{E}_{\mathbf{F}_{L-1}|\mathbf{F}_{L-2}}\left[\mathbb{E}_{\mathbf{f}_L|\mathbf{F}_{L-1}}\left[\exp(i\mathbf{t}^\top \mathbf{f}_L)\right]\right]\right]\right]$$

$$\text{(law of total expectation)}$$

$$= \mathbb{E}_{\mathbf{F}_1}\left[\mathbb{E}_{\mathbf{F}_2|\mathbf{F}_1}\left[\ldots \mathbb{E}_{\mathbf{F}_{L-1}|\mathbf{F}_{L-2}}\left[\exp\left(-\frac{1}{2}\mathbf{t}^\top \mathbf{K}_L\left(\mathbf{F}_{L-1}, \mathbf{F}_{L-1}\right)\mathbf{t}\right)\right]\right]\right]$$

$$\text{(Gaussian characteristic function)}$$

$$\geq \exp\left(-\frac{1}{2}\mathbf{t}^\top \mathbb{E}_{\mathbf{F}_1}\left[\mathbb{E}_{\mathbf{F}_2|\mathbf{F}_1}\left[\ldots \mathbb{E}_{\mathbf{F}_{L-1}|\mathbf{F}_{L-2}}\left[\mathbf{K}_L\left(\mathbf{F}_{L-1}, \mathbf{F}_{L-1}\right)\right]\right]\right]\mathbf{t}\right)$$

$$\text{(Jensen's inequality, strict convexity of exp)}$$

$$= \exp\left(-\frac{1}{2}\mathbf{t}^\top \mathbb{E}_{\mathbf{F}_{L-1}}\left[\mathbf{K}_L\left(\mathbf{F}_{L-1}, \mathbf{F}_{L-1}\right)\right]\mathbf{t}\right) \qquad \text{(law of total expectation)}$$

$$= \exp\left(-\frac{1}{2}\mathbf{t}^\top \mathbb{E}_{\mathbf{f}_L}\left[\mathbf{f}_L \mathbf{f}_L^\top\right]\mathbf{t}\right) = \exp\left(-\frac{1}{2}\mathbf{t}^\top \mathbf{K}_{\text{lim}}\mathbf{t}\right) = \mathbb{E}_{\mathbf{g}\sim\mathcal{N}(\mathbf{0},\mathbf{K}_{\text{lim}})}\left[\exp(i\mathbf{t}^\top \mathbf{g})\right].$$

$$(33)$$

Thus, we have

$$p(\mathbf{f}_L = \mathbf{0}) = \int \mathbb{E}_{\mathbf{f}_L}\left[\exp(i\mathbf{t}^\top \mathbf{f}_L)\right]\,\mathrm{d}\mathbf{t}$$

$$\geq \int \mathbb{E}_{\mathbf{g}\sim\mathcal{N}(\mathbf{0},\mathbf{K}_{\text{lim}})}\left[\exp(i\mathbf{t}^\top \mathbf{g})\right]\,\mathrm{d}\mathbf{t} = \mathcal{N}\left(\mathbf{g} = \mathbf{0}; \mathbf{0}, \mathbf{K}_{\text{lim}}\right),$$

which completes the proof. $\qquad\square$

It is worth noting that the Jensen gap of the characteristic function cascades with depth. For example, given the 3-layer model $f_3(\mathbf{f}_2(\mathbf{f}_1(\cdot)))$, we have:

$$\underbrace{\mathbb{E}_{\mathbf{F}_1}\left[\mathbb{E}_{\mathbf{F}_2|\mathbf{F}_1}\left[\exp\left(-\frac{1}{2}\mathbf{t}^\top \mathbf{K}_3\mathbf{t}\right)\right]\right]}_{\text{CF of 3-layer Deep GP marginal}} \geq \underbrace{\mathbb{E}_{\mathbf{F}_1}\left[\exp\left(-\frac{1}{2}\mathbf{t}^\top \mathbb{E}_{\mathbf{F}_2|\mathbf{F}_1}\left[\mathbf{K}_3\right]\mathbf{t}\right)\right]}_{\text{CF of 2-layer Deep GP marginal}}$$

$$\geq \underbrace{\exp\left(-\frac{1}{2}\mathbf{t}^\top \mathbb{E}_{\mathbf{F}_1}\left[\mathbb{E}_{\mathbf{F}_2|\mathbf{F}_1}\left[\mathbf{K}_3\right]\right]\mathbf{t}\right)}_{\text{CF of } \mathcal{N}(\mathbf{0}, \mathbb{E}_{\mathbf{F}_1}[\mathbb{E}_{\mathbf{F}_2|\mathbf{F}_1}[\mathbf{K}_3]])},$$

Consequentially, the peaks of deeper models will be sharper than those of shallower models. This is analogous to the tail effects of depth analyzed in Sec. 5.

**Theorem 3 (Restated).** *Let $\mathbf{t} \in \mathbb{R}^N$. Using the same setup, notation, and assumptions as Thm. 2, the odd moments of $\mathbf{t}^\top \mathbf{f}_L$ are zero and the even moments larger than 2 are super-Gaussian, i.e. $\mathbb{E}_{\mathbf{f}_L}[(\mathbf{t}^\top \mathbf{f}_L)^r] \geq \mathbb{E}_{\mathbf{g}\sim\mathcal{N}(\mathbf{0},\mathbf{K}_{lim})}[(\mathbf{t}^\top \mathbf{g})^r]$ for all even $r \geq 4$. Moreover, if $k_L(\cdot, \cdot)$ is bounded almost everywhere, the moment generating function $\mathbb{E}_{\mathbf{f}_L}[\exp(\mathbf{t}^\top \mathbf{f}_L)]$ exists and is similarly super-Gaussian.*

*Proof.* We can express the moments of $\mathbf{t}^\top \mathbf{f}_L$ as:

$$\mathbb{E}_{\mathbf{f}_L}\left[(\mathbf{t}^\top \mathbf{f}_L)^r\right] = \mathbb{E}_{\mathbf{F}_1}\left[\mathbb{E}_{\mathbf{F}_2|\mathbf{F}_1}\left[\ldots \mathbb{E}_{\mathbf{F}_{L-1}|\mathbf{F}_{L-2}}\left[\mathbb{E}_{\mathbf{f}_L|\mathbf{F}_{L-1}}\left[(\mathbf{t}^\top \mathbf{f}_L)^r\right]\right]\right]\right],$$

and note that the innermost expectation can be simplified to:

$$\mathbb{E}_{\mathbf{f}_L|\mathbf{F}_{L-1}}\left[(\mathbf{t}^\top \mathbf{f}_L)^r\right] = \begin{cases} 0 & r \text{ is odd} \\ \left(\mathbf{t}^\top \mathbf{K}_L(\mathbf{F}_{L-1}, \mathbf{F}_{L-1})\mathbf{t}\right)^{\frac{r}{2}}(r-1)!! & r \text{ is even.} \end{cases} \quad \text{(Gaussian moments)}$$

For odd $r$, note that this implies $\mathbb{E}_{\mathbf{f}_L}[(\mathbf{t}^\top \mathbf{f}_L)^r] = 0$. For even $r \geq 4$, note that $\mathbf{t}^\top \mathbf{K}_L(\mathbf{F}_{L-1}, \mathbf{F}_{L-1})\mathbf{t} \geq 0$ by positive definiteness of kernels, and note that $(z)^{r/2}$ is convex for all $z \geq 0$. Following the same logic as in the proof for Thm. 2, we have:

$$
\mathbb{E}_{\mathbf{f}_L}\left[\left(\mathbf{t}^\top \mathbf{f}_L\right)^r\right] = \mathbb{E}_{\mathbf{F}_1}\left[\mathbb{E}_{\mathbf{F}_2|\mathbf{F}_1}\left[\cdots \mathbb{E}_{\mathbf{F}_{L-1}|\mathbf{F}_{L-2}}\left[\left(\mathbf{t}^\top \mathbf{K}_L(\mathbf{F}_{L-1}, \mathbf{F}_{L-1})\mathbf{t}\right)^{\frac{r}{2}}(r-1)!!\right]\right]\right]
$$

$$
\geq \left(\mathbf{t}^\top \mathbb{E}_{\mathbf{F}_1}\left[\mathbb{E}_{\mathbf{F}_2|\mathbf{F}_1}\left[\cdots \mathbb{E}_{\mathbf{F}_{L-1}|\mathbf{F}_{L-2}}[\mathbf{K}_L(\mathbf{F}_{L-1}, \mathbf{F}_{L-1})]\right]\right]\mathbf{t}\right)^{\frac{r}{2}}(r-1)!!
$$
$$
\text{(Jensen's inequality, strict convexity)}
$$

$$
= \left(\mathbf{t}^\top \mathbf{K}_{\text{lim}}\mathbf{t}\right)^{\frac{r}{2}}(r-1)!! = \mathbb{E}_{\mathcal{N}(\mathbf{g};\mathbf{0},\mathbf{K}_{\text{DGP}}(\mathbf{X},\mathbf{X}))}\left[(\mathbf{t}^\top \mathbf{g})^r\right]
$$

A similar proof will show that the moment generating function is similarly super-Gaussian:

$$
\mathbb{E}_{\mathbf{f}_L}\left[\exp(\mathbf{t}^\top \mathbf{f}_L)\right] = \mathbb{E}_{\mathbf{F}_{L-1}}\left[\exp\left(\frac{1}{2}\mathbf{t}^\top \mathbf{K}_L\left(\mathbf{F}_{L-1}, \mathbf{F}_{L-1}\right)\mathbf{t}\right)\right] \tag{34}
$$

$$
= \mathbb{E}_{\mathbf{F}_1}\left[\mathbb{E}_{\mathbf{F}_2|\mathbf{F}_1}\left[\cdots \mathbb{E}_{\mathbf{F}_{L-1}|\mathbf{F}_{L-2}}\left[\exp\left(\frac{1}{2}\mathbf{t}^\top \mathbf{K}_L\left(\mathbf{F}_{L-1}, \mathbf{F}_{L-1}\right)\mathbf{t}\right)\right]\right]\right]
$$

$$
\geq \exp\left(\frac{1}{2}\mathbf{t}^\top \mathbb{E}_{\mathbf{F}_1}\left[\mathbb{E}_{\mathbf{F}_2|\mathbf{F}_1}\left[\cdots \mathbb{E}_{\mathbf{F}_{L-1}|\mathbf{F}_{L-2}}[\mathbf{K}_L\left(\mathbf{F}_{L-1}, \mathbf{F}_{L-1}\right)]\right]\right]\mathbf{t}\right)
$$

$$
= \exp\left(\frac{1}{2}\mathbf{t}^\top \mathbf{K}_{\text{lim}}\mathbf{t}\right) = \mathbb{E}_{\mathcal{N}(\mathbf{g};\mathbf{0},\mathbf{K}_{\text{lim}})}\left[\exp(\mathbf{g}^\top \mathbf{t})\right].
$$

We know that the moment generating function exists because, by assumption, $k_L(\cdot,\cdot)$ is bounded almost everywhere, and thus the integral defined by the expectation in Eq. (34) is finite. $\qquad\square$

## G   Derivation of Deep GP Covariances and Limiting GP Covariances

Here we derive the prior covariances of various Deep GP architectures, as well as the covariances of their corresponding infinite width GP limits.

### G.1   RBF + Additive RBF

First, consider a two layer Deep GP $f_2(\mathbf{f}_1(\cdot))$ where the first layer uses a RBF covariance and the second layer uses a sum of 1-dimensional RBF covariances:

$$
k_1(\mathbf{x}, \mathbf{x}') = o_1^2 \exp\left(\frac{-\|\mathbf{x} - \mathbf{x}'\|_2^2}{2\ell_1^2}\right),
$$

$$
k_2(\mathbf{f}_1(\mathbf{x}), \mathbf{f}_1(\mathbf{x}')) = \frac{o_2^2}{H_1}\sum_{i=1}^{H_1}\exp\left(-\frac{(f_1^{(i)}(\mathbf{x}) - f_1^{(i)}(\mathbf{x}'))^2}{2\ell_2^2}\right),
$$

where $H_1$ is the width of the Deep GP, and $o_1$, $\ell_1$, $o_2$, and $\ell_2$ are hyperparameters. This is the Deep GP architecture most commonly explored in this paper.

To calculate the covariance between $f_2(\mathbf{f}_1(\mathbf{x}))$ and $f_2(\mathbf{f}_1(\mathbf{x}'))$, we first note that $\tau_i \triangleq f_1^{(i)}(\mathbf{x}) - f_1^{(i)}(\mathbf{x}')$ is Gaussian distributed:

$$
\tau_i = f_1^{(i)}(\mathbf{x}) - f_1^{(i)}(\mathbf{x}') \sim \mathcal{N}\left(0, \sigma^2\right), \qquad \sigma^2 \triangleq k_1(\mathbf{x}, \mathbf{x}) + k_1(\mathbf{x}', \mathbf{x}') - 2k_1(\mathbf{x}, \mathbf{x}'). \tag{35}
$$

The covariance of the Deep GP is therefore given as:

$$\underset{\text{RBF+add-RBF}}{\mathbb{E}}[f_2(\mathbf{f}_1(\mathbf{x}))f_2(\mathbf{f}_1(\mathbf{x}'))] = \underset{\mathbf{f}_1(\mathbf{x}),\mathbf{f}_1(\mathbf{x}')}{\mathbb{E}}[k_2(\mathbf{f}_1(\mathbf{x}),\mathbf{f}_1(\mathbf{x}')]$$

$$= \underset{\tau_i}{\mathbb{E}}\left[\frac{o_2^2}{H_1}\sum_{i=1}^{H_1}\exp\left(-\frac{\tau_i^2}{2\ell_2^2}\right)\right] \qquad (\tau_i \triangleq f_1^{(i)}(\mathbf{x}) - f_1^{(i)}(\mathbf{x}'))$$

$$= o_2^2 \underset{\tau_1}{\mathbb{E}}\left[\exp\left(-\frac{\tau_1^2}{2\ell_2^2}\right)\right] \qquad (\tau_i \text{ are i.i.d.})$$

$$= \frac{o_2^2}{\sqrt{2\pi\sigma^2}}\int_{-\infty}^{\infty}\exp\left(\frac{-\tau_1^2}{2\ell_2^2}\right)\exp\left(\frac{-\tau_1^2}{2\sigma^2}\right)\,\mathrm{d}\tau_1$$

$$= \frac{o_2^2}{\sqrt{2\pi\sigma^2}}\int_{-\infty}^{\infty}\exp\left(-\tau_1^2\frac{\sigma^2+\ell_2^2}{2\sigma^2\ell_2^2}\right)\,\mathrm{d}\tau_1$$

$$= \frac{o_2^2}{\sqrt{2\pi\sigma^2}}\sqrt{\frac{2\pi\sigma^2\ell_2^2}{\sigma^2+\ell_2^2}}. \qquad \text{(Gaussian normalizing constant)}$$

Plugging in $\sigma^2$ from Eq. (35), we have

$$\underset{\text{RBF+add-RBF}}{\mathbb{E}}[f_2(\mathbf{f}_1(\mathbf{x}))f_2(\mathbf{f}_1(\mathbf{x}'))] = o_2^2\left(1 + \frac{2o_1^2\left(1-\exp\left(-\frac{\|\mathbf{x}-\mathbf{x}'\|_2^2}{2\ell_1^2}\right)\right)}{\ell_2^2}\right)^{-1/2}. \tag{36}$$

Note that this covariance is the same, regardless of the Deep GP width $H_1$. Therefore, it is also the covariance of the infinite width GP limit. More generally, if we replace the first RBF covariance with an arbitrary covariance function $k_1(\cdot,\cdot)$ we have:

$$\underset{k_1\text{+add-RBF}}{\mathbb{E}}[f_2(\mathbf{f}_1(\mathbf{x}))f_2(\mathbf{f}_1(\mathbf{x}'))] = o_2^2\left(1 + \frac{k_1(\mathbf{x},\mathbf{x}) + k_1(\mathbf{x}',\mathbf{x}') - 2k_1(\mathbf{x},\mathbf{x}')}{\ell_2^2}\right)^{-1/2}. \tag{37}$$

A similar derivation can be found in [62].

## G.2   RBF + (Non-Additive) RBF

Now consider a two layer Deep GP $f_2(\mathbf{f}_1(\cdot))$ where the first and second layers both use *non-additive* RBF covariance functions:

$$k_1(\mathbf{x},\mathbf{x}') = o_1^2\exp\left(\frac{-\|\mathbf{x}-\mathbf{x}'\|_2^2}{2\ell_1^2}\right),$$

$$k_2(\mathbf{f}_1(\mathbf{x}),\mathbf{f}_1(\mathbf{x}')) = o_2^2\exp\left(-\frac{\|\mathbf{f}_1(\mathbf{x})-\mathbf{f}_1(\mathbf{x}')\|_2^2}{2H_1\ell_2^2}\right),$$

where the $1/H_1$ factor is included to reduce the impact of the dimensionality of $\mathbf{f}_1(\cdot)$. This is a very common Deep GP architecture [e.g. 19, 24, 27, 79], and it is the architecture in the Sec. 4 example. Crucially, the RBF kernel decomposes as a product across its dimensions:

$$k_2(\mathbf{f}_1(\mathbf{x}),\mathbf{f}_1(\mathbf{x}')) = o_2^2\exp\left(-\frac{\|\mathbf{f}_1(\mathbf{x})-\mathbf{f}_1(\mathbf{x}')\|_2^2}{2H_1\ell_2^2}\right) = o_2^2\prod_{i=1}^{H_1}\exp\left(-\frac{\left(f_1^{(i)}(\mathbf{x})-f_1^{(i)}(\mathbf{x}')\right)^2}{2H_1\ell_2^2}\right)$$

Since the $f_1^{(i)}(\cdot)$ are independent, we have:

$$\mathop{\mathbb{E}}_{\text{RBF+RBF}}[f_2(\mathbf{f}_1(\mathbf{x}))f_2(\mathbf{f}_1(\mathbf{x}'))] = \mathop{\mathbb{E}}_{\mathbf{f}_1(\mathbf{x}),\mathbf{f}_1(\mathbf{x}')}\left[o_2^2\prod_{i=1}^{H_1}\exp\left(-\frac{\left(f_1^{(i)}(\mathbf{x})-f_1^{(i)}(\mathbf{x}')\right)^2}{2H_1\ell_2^2}\right)\right]$$

$$= o_2^2\prod_{i=1}^{H_1}\mathop{\mathbb{E}}_{f_1^{(i)}(\mathbf{x}),f_1^{(i)}(\mathbf{x}')}\left[\exp\left(-\frac{\left(f_1^{(i)}(\mathbf{x})-f_1^{(i)}(\mathbf{x}')\right)^2}{2H_1\ell_2^2}\right)\right]$$

$$= o_2^2\prod_{i=1}^{H_1}\mathop{\mathbb{E}}_{\text{RBF+add-RBF}}\left[f_2\left(\frac{f_1^{(1)}(\mathbf{x})}{\sqrt{H_1}}\right)f_2\left(\frac{f_1^{(1)}(\mathbf{x}')}{\sqrt{H_1}}\right)\right],$$

Plugging in Eq. (37), we have

$$\mathop{\mathbb{E}}_{\text{RBF+RBF}}[f_2(\mathbf{f}_1(\mathbf{x}))f_2(\mathbf{f}_1(\mathbf{x}'))] = o_2^2\left(1+\frac{k_1(\mathbf{x},\mathbf{x})+k_1(\mathbf{x}',\mathbf{x}')-2k_1(\mathbf{x},\mathbf{x}')}{H_1\ell_2^2}\right)^{-H_1/2}.$$

$$= o_2^2\left(1+\frac{2o_1^2\left(1-\exp\left(-\frac{\|\mathbf{x}-\mathbf{x}'\|_2^2}{2\ell_1^2}\right)\right)}{H_1\ell_2^2}\right)^{-H_1/2}. \qquad (38)$$

In the limit as $H_1\to\infty$, this second moment becomes:

$$\lim_{H_1\to\infty}\mathop{\mathbb{E}}_{\text{RBF+RBF}}[f_2(\mathbf{f}_1(\mathbf{x}))f_2(\mathbf{f}_1(\mathbf{x}'))] = o_2^2\exp\left(-\frac{k_1(\mathbf{x},\mathbf{x})+k_1(\mathbf{x}',\mathbf{x}')-2k_1(\mathbf{x},\mathbf{x}')}{2\ell_2^2}\right).$$

$$= o_2^2\exp\left(\frac{o_1^2}{\ell_2^2}\exp\left(-\frac{\|\mathbf{x}-\mathbf{x}'\|_2^2}{2\ell_1^2}\right)-1\right). \qquad (39)$$

### G.3   RBF + Additive RBF + Additive RBF

Now consider the three layer Deep GP $f_3(\mathbf{f}_2(\mathbf{f}_1(\cdot)))$, where the first layer uses an RBF covariance and the other layers use sums of 1-dimensional RBF covariances.

$$k_1(\mathbf{x},\mathbf{x}') = o_1^2\exp\left(\frac{-\|\mathbf{x}-\mathbf{x}'\|_2^2}{2\ell_1^2}\right),$$

$$k_2(\mathbf{f}_1(\mathbf{x}),\mathbf{f}_1(\mathbf{x}')) = \frac{o_2^2}{H_1}\sum_{i=1}^{H_1}\exp\left(-\frac{(f_1^{(i)}(\mathbf{x})-f_1^{(i)}(\mathbf{x}'))^2}{2\ell_2^2}\right),$$

$$k_3(\mathbf{f}_2(\mathbf{f}_1(\mathbf{x})),\mathbf{f}_2(\mathbf{f}_1(\mathbf{x}'))) = \frac{o_3^2}{H_2}\sum_{i=1}^{H_2}\exp\left(-\frac{\left(f_2^{(i)}(\mathbf{f}_1(\mathbf{x}))-f_2^{(i)}(\mathbf{f}_1(\mathbf{x}'))\right)^2}{2\ell_3^2}\right),$$

where $H_1$, $H_2$ are the widths of the first and second layers, and $o_1$, $\ell_1$, $o_2$, $\ell_2$, $o_3$ and $\ell_3$ are hyperparameters. We use this architecture in Sec. 5 and Sec. 6.1.

Unfortunately, it is intractable to compute the second moment of this Deep GP in closed form. To see why this is the case, note that:

$$\mathop{\mathbb{E}}_{\text{RBF+add-RBF+add-RBF}}[f_3(\mathbf{f}_2(\mathbf{f}_1(\mathbf{x})))\ f_3(\mathbf{f}_2(\mathbf{f}_1(\mathbf{x}')))] = \mathop{\mathbb{E}}_{\mathbf{f}_2(\mathbf{f}_1(\mathbf{x})),\mathbf{f}_2(\mathbf{f}_1(\mathbf{x}'))}[k_3(\mathbf{f}_2(\mathbf{f}_1(\mathbf{x})),\ \mathbf{f}_2(\mathbf{f}_1(\mathbf{x}')))]$$
$$\qquad (40)$$

In other words, computing the covariance requires taking the expectation over the Deep GP marginal $\mathbf{f}_2(\mathbf{f}_1(\mathbf{x})),\mathbf{f}_2(\mathbf{f}_1(\mathbf{x}'))$ which is intractable to compute. We do note that we can approximate this marginal with Gauss-Hermite quadrature if $H_1$ is sufficiently small. Moreover, unlike the 2-layer case, the width $H_1$ affects the second moment of this 3-layer Deep GP. This is because changing $H_1$ changes the marginal distribution $\mathbf{f}_2(\mathbf{f}_1(\mathbf{x})),\mathbf{f}_2(\mathbf{f}_1(\mathbf{x}'))$, which ultimately impacts the expectation in Eq. (40). Changing the value of $H_2$ does not affect the covariance by linearity of expectation, assuming that we hold $H_1$ constant.

As $H_1 \to \infty$ and $\mathbf{f}_2(\mathbf{f}_1(\cdot))$ converges to a Gaussian process, the expectation in Eq. (40) becomes tractable again. $f_3(\mathbf{f}_2(\mathbf{f}_1(\cdot)))$ effectively becomes a 2-layer Deep GP, where the first layer has covariance given by Eq. (36) and the second layer is the sum of 1-dimensional RBF covariances. Thus, combining Eq. (36) and Eq. (37), we can compute the covariance of the limiting GP:

$$
\lim_{\substack{H_2 \to \infty}} H_1 \to \infty \quad \underset{\text{RBF+add-RBF+add-RBF}}{\mathbb{E}} \left[ f_3(\mathbf{f}_2(\mathbf{f}_1(\mathbf{x}))) \; f_3(\mathbf{f}_2(\mathbf{f}_1(\mathbf{x}'))) \right]
$$

$$
= o_3^2 \left( 1 + \frac{2 o_2 \left( 1 - \left( 1 + \frac{2 o_1^2 \left( 1 - \exp\left( -\frac{\|\mathbf{x}-\mathbf{x}'\|_2^2}{2\ell_1^2} \right) \right)}{\ell_2^2} \right)^{-1/2} \right)}{\ell_3^2} \right)^{-1/2}
$$

A similar derivation can be found in [62].

### G.4  Neural networks

The prior second moment of a neural network with ReLU activations and a single hidden layer (i.e. the construction in Eq. 1) is given by the arc-cosine kernel [23]:

$$
\underset{\text{2-layer NN}}{\mathbb{E}} \left[ f_2(\mathbf{f}_1(\mathbf{x})) f_2(\mathbf{f}_1(\mathbf{x}')) \right] = \beta^2 + \frac{1}{2\pi} \|\mathbf{x}\| \, \|\mathbf{x}'\| \left( \sin(\theta) + (\pi - \theta) \cos(\theta) \right), \qquad (41)
$$

where $\theta = \cos^{-1}\left( (\mathbf{x}^\top \mathbf{x}')/(\|\mathbf{x}\| \, \|\mathbf{x}'\|) \right)$. Note that this covariance is constant regardless of $H_1$.

As with the 3-layer RBF Deep GP, we cannot compute the prior second moment of deeper neural networks in closed form. Moreover, once neural networks have more than 1 hidden layer, then the width of hidden layers affects the covariance. Nevertheless, we can compute the limiting infinite width covariance using the recursive formula defined in [23, 56].

## H  Experimental Details

The experiments are implemented in PyTorch [73], supplemented by the Pyro [15] and GPyTorch [37] libraries – all of which are open source. We ran them on a cluster with GTX1080 and GTX2080 GPU, and we estimate that we used $\approx 1,000$ hours of GPU compute time. The largest experiments (CIFAR10 ResNets with maximum width) require $48GB$ of GPU memory; all other experiments only require $\leq 11GB$ of memory.

**Datasets.** The datasets for the regression experiments are from the UCI repository [10]. Unless otherwise stated, we split these datasets into $75\%$ training data, $15\%$ test data, and $10\%$ validation data. For larger datasets, we subsample the training dataset to a maximum of $N = 1,000$ data points. All input features are normalized to be between $-1$ and $1$, and the $\mathbf{y}$ values are z-scored to have $0$ mean and unit variance. For the non-Bayesian neural network experiments, we use the MNIST [55] and CIFAR10 [54] datasets. We z-score the inputs so that each channel has $0$ mean and unit variance. We use the standard $10,000$ data point test sets, and subsample the remaining data for training. Models are trained without data augmentation.

**Deep GP models.** All Deep GP models use GP layers with zero prior mean. We perform inference without making any scalable approximations, though we do add a constant diagonal of $10^{-4}$ to all prior covariances for stability. We perform inference using the NUTS sampler [48] implemented in Pyro [15], using 500 warmup steps, drawing 500 samples, a target acceptance probability of 0.8, an initial learning rate of 0.1, and a maximum tree depth of 10. To improve inference, we infer the "whitened" latent variables $\mathbf{L}_1^{-1}\mathbf{F}_1$ and $\mathbf{L}_2^{-1}\mathbf{f}_2$, where $\mathbf{L}_1$ and $\mathbf{L}_2$ are the Cholesky factors of $\mathbf{K}_1(\mathbf{X}, \mathbf{X})$ and $\mathbf{K}_2(\mathbf{F}_1, \mathbf{F}_1)$ respectively. For all width $\geq 2$ Deep GP, we initialize the latent variables by running $1,000$ steps of Adam [53] with learning rate 0.01 on the maximum a posteriori Deep GP objective. Because width-1 Deep GP inference is more challenging (see the control experiment

in Appx. D), we instead initialize the latent variables of these models from the mean of a doubly stochastic variational Deep GP [79], where we use 300 inducing points per layer, 10 function samples, and a minibatch size of 128. We optimize the variational Deep GP with Adam for 2,000 iterations, using an initial learning rate of 0.01, dropping it by a factor of 10 after 50% and 75% of training.

For each Deep GP model, we use hyperparameters that maximize the log marginal likelihood of the corresponding limiting GP. To find these hyperparameters, we perform 100 iterations of Adam on the limiting GP using a learning rate of 0.1, initializing all covariance hyperparameters to 1 and initializing the likelihood observational noise to 0.2.

**Bayesian neural network models.** We train the Bayesian neural networks in a very similar manner. However, we perform 2,000 warmup steps and draw 1,000 samples using NUTS. Again, we use the same hyperparameters as the optimized limiting GP.

**MNIST experiments (Non-Bayesian neural networks).** We train 3-layer (2 hidden-layer) MLP with an L2 regularization constant of $\in \{10^{-5}, 10^{-6}, 10^{-7}, 10^{-8}\}$, which corresponds to per-parameter priors of $\{\mathcal{N}(0, 2), \mathcal{N}(0, 20), \mathcal{N}(0, 200), \mathcal{N}(0, 2000)\}$ when $N = 50,000$. Following Eq. (1), the output of layer $\ell$ is scaled by a factor of $1/\sqrt{H_{\ell-1}}$, where $H_{\ell-1}$ is the width of layer $\ell - 1$. We train the models for 20,000 iterations using the Adam optimizer. Following Lee et al. [56], we perform a random search over the learning rate and batch size parameters. More specifically, we randomly choose 10 learning rate/batch size tuples from $[0.001, 0.2] \times \{16, 32, 64, 128, 256\}$, and select the hyperparameters that generate the best accuracy on the withheld training data. Note that these hyperparameters do not affect the prior of the model's Bayesian analog; they only impact the optimization dynamics. We drop the learning rate by a factor of 10 after 50% and 75% of training.

**CIFAR10 experiments (Non-Bayesian neural networks).** We do not perform the $1/\sqrt{H_{\ell-1}}$ scaling on the ResNet models, as this scaling is undone by batch normalization. Models are optimized using SGD with an initial learning rate of 0.1, following the same schedule as the MNIST models. The L2 regularization constant ($10^{-4}$) corresponds to a per-parameter prior of $\mathcal{N}(0, 0.2)$. Note that these hyperparameters exactly match those suggested by He et al. [47] and Zagoruyko and Komodakis [92]. We train each model for 40,000 iterations with a minibatch of 256.

**Effect of depth experiments in Sec. 6.1.** In these experiments, our goal is to investigate the effects of depth while controlling for the first and second moments of the Deep GP models. To that end, we construct a 3-layer Deep GP, 2-layer Deep GP, and a single-layer GP all with zero mean and the same prior covariance. The 3-layer Deep GP uses a RBF covariance in the first layer, and sums of 1-dimensional RBF kernels in the other two layers. We set the widths to be $H_1 = 2$ and $H_2 = 8$. The 2-layer Deep GP uses a RBF covariance in the first layer with a width of $H_1 = 2$, while the second layer uses the following covariance:

$$o_3^2 \left( 1 + \frac{2o_2^2 \left( 1 - \frac{1}{H_1} \sum_{i=1}^{H_1} \exp\left( -\left( f_1^{(i)}(\mathbf{x}) - f_1^{(i)}(\mathbf{x}') \right)^2 / (2\ell_2) \right) \right)}{\ell_3^2} \right)^{-1/2} . \tag{42}$$

For the single layer GP, we compute the covariance of the 3-layer model using the formula in Eq. (40). We approximate the marginal distribution $p(\mathbf{f}_2(\mathbf{f}_1(\mathbf{x})), \mathbf{f}_2(\mathbf{f}_1(\mathbf{x}')))$ using Gauss-Hermite quadrature with 11 nodes. We empirically confirm that the single-layer, 2-layer, and 3-layer models have the same prior covariance.

**Visualizing $N = 2$ marginal densities in Sec. 5.** The 3-layer Deep GP uses a RBF covariance in the first layer, and sums of 1-dimensional RBF kernels in the other two layers. We vary both widths $H_1$ and $H_2$ simultaneously, and we set all hyperparameters to 1. The 2-layer Deep GP are designed to match the prior covariance of the width = 1 3-layer model. To that end, the first layer uses an RBF covariance, and the second layer uses the following covariance:

$$\left( 1 + 2 \left( 1 - \sum_{i=1}^{H_1} \exp\left( \frac{-\left( f_1^{(i)}(\mathbf{x}) - f_1^{(i)}(\mathbf{x}') \right)^2}{2} \right) \right) \right)^{-1/2} . \tag{43}$$

Again, we empirically verify that these models have the same prior covariance. We approximate the marginal densities at 400 evenly spaced grid points on $\mathbf{y} \in [-3, 3] \times [3, 3]$ using Gauss-Hermite quadrature with 7 nodes.