# OpenReview forum: "The Limitations of Large Width in Neural Networks: A Deep Gaussian Process Perspective"
_NeurIPS.cc/2021/Conference — NeurIPS 2021 Poster_

### Official Review · Reviewer_PVcF · 2021-07-13

**Rating:** 6
**Confidence:** 3

**Summary:**

The paper argues that deep Gaussian Processes with large widths and more than 2 layers have undesirable properties. In particular, the paper shows that firstly, deep GP prior in the GP limit is essentially shallow, and secondly, deep GP posterior then loses data adaptivity in the hidden layers. The paper further examines the role of depth and argues that the deep GP prior, if not equal to a GP, has heavy-tailed behaviors.

==================================
Post-rebuttal:

Thanks for the reply and clarification!

While I agree that very large width hurts in the context of deep GP as shown in the paper, I also agree with other reviewers that the message that very large width hurts generally is overly strong. The literature has a mixed bag of results concerning widths -- theoretically or empirically, so a general statement should be validated very carefully.

I'd like to keep my score.

**Limitations And Societal Impact:**

Yes

**Main Review:**

The subject of the paper is timely, and the conclusions are interesting. In a way, it is not surprising that the deep GP prior becomes shallow in the GP limit; specifically Lemma 1 is expectable since intuitively a mixture of zero-mean Gaussians, if equal to a Gaussian, should comprise of same Gaussian components. (To be fair, Lemma 2 is more general than this.) What is interesting in the paper is the interpretation that in the hidden layers, a certain hierarchical structure is lost and hence so is data adaptivity in the posterior. This message is a strong and noteworthy one.

Yet this important message about the posterior is where I do not find it satisfying yet. In particular, the argument around (6) potentially ignores some important limit exchange issue, technically speaking. The conceptual problem I have in mind is the following. It is true that in the prior the kernel converges pointwise to a degenerate limit as the width increases. However this doesn’t exclude the possibility that in the process of computing the posterior (or in general, a process that involves fitting the training data), the movement of the weight of the first layer explodes with large width. In such scenario, it is unclear if in the posterior, the kernel can still be close to the degenerate limit (in an appropriate sense such that (6) holds).

Another issue — of less importance in my personal opinion — is that of the experimental results in Section 6.2. This section aims to argue against large widths in contexts beyond GP. However the way I read the results in Figure 4, they say that the performance does not necessarily increase monotonically with width. I feel even more mixed when contrasting this figure with other papers, such as [51]. The notion of “sufficient capacity” is not defined, nor is it illustrated numerically. I have a similar concern regarding Table 1, where the message that deeper is better is not clear.

**Time Spent Reviewing:**

8

---

> ### Author Response · Authors · 2021-08-09
> **Response to PVcF**
>
> We are pleased that you find our paper timely and interesting, and we appreciate your useful comments. We hope to address your major concerns below. Per your comments on the Figure 4 results, please refer to our discussions with Reviewers zKHo and h2Yp.
>
> > “It is true that in the prior the kernel converges pointwise to a degenerate limit as the width increases. However this doesn’t exclude the possibility that in the process of computing the posterior the movement of the weight of the first layer explodes with large width.”
>
> You bring up an interesting point that warrants discussion in the final paper; however, we are confident in the correctness of our posterior analysis. Prior work theoretically confirms that convergence to a GP prior indeed implies posterior convergence to the corresponding GP’s posterior (see  [Hron et al. (2020), Proposition 1](https://arxiv.org/pdf/2006.10541) - a reference that we will add to Section 4). The RHS of Eq. 6 is the posterior distribution of the limiting GP, and thus it is the posterior of the infinite width Deep GP. It is true that computing or sampling from this posterior may encounter computational or optimization challenges, such as the ones that you mention; however the true posterior distribution is agnostic to the inference procedure and therefore these concerns do not affect its distributional form.
>
> > “The notion of “sufficient capacity” is not defined, nor is it illustrated numerically.“
>
> This is a fair critique, and one which we will address more formally in the final version. Under any standard definition of capacity, such as VC dimension, neural networks have finite capacity whereas nonparametric Deep GP (with universal covariance functions) have infinite capacity (see Appendix B). Our theory in Sections 3-5 suggests that neural network width controls a capacity/adaptability tradeoff (analogous to other classic machine learning tradeoffs). While the optimal capacity of a NN depends on several hard-to-measure factors and dataset-dependent features, it stands to reason that -- after sufficient width -- additional neurons make the prior distribution increasingly Gaussian while offering little additional gains in modeling precision (as suggested by the LL optima in Figure 3, orange line). This is the regime that we refer to as “sufficient capacity.”

---

### Official Review · Reviewer_zKHo · 2021-07-15

**Rating:** 6
**Confidence:** 3

**Summary:**

The paper shows that in the limit of infinite width a Deep Gaussian Process (DGP) converges to an ordinary Gaussian Process (GP). Indeed, in the limit of infinite width the covariance function of each hidden GP layer becomes deterministic (as a consequence of the strong law of large numbers). This allows proving that the marginal prior covariance  of the DGP becomes gaussian (Theorem1). The  consequence of this theorem is that as the width increases DGP models lose their ability to adapt the hidden covariance functions to the input data (section 4).
This theoretical result is supported by numerical experiments (section 6.1).
The paper also addresses the convergence of a DGP to a GP with increasing depth and width (section 5).
Since it is known that deep neural networks (DNN) approach a GP as they are made wider and wider,  Theorem 1 is used to  claim that a very large width can be detrimental to the expressivity of a DNN (and not only a DGP).



**Limitations And Societal Impact:**

In my opinion the limitations are not sufficiently discussed, in particular concerning  the connection of the results to the theory of DNNs.

**Main Review:**

The paper is  well written and the proofs of the theorems/lemmas are, to my understanding, correct. The connections with the relevant literature are also discussed, with an important exception (see below).

I have a concern related to the meaningfulness of the extension of the results to standard neural networks.  The test accuracy reported in Figure 4, rightmost panel, is significantly lower than the accuracy obtained with non-Bayesian approaches on CIFAR10. (see for example https://paperswithcode.com/sota/image-classification-on-cifar-10?tag_filter=3) This is for sure partially due to the lack of data augmentation (which is acceptable in a numerical experiment) but, in my opinion, also to the training protocol. If I understood correctly, the same weight decay is applied for every model. What would happen if one would tune this decay optimally? Maybe, as stated by the authors “.. by changing the weight decay parameter, may lessen the consequences of width that we observe” (line 322-323). Indeed, setting the weight decay at initialization amounts simply  to set the inverse variance of the gaussian prior on the network weights. It is well known empirically that smaller models require larger weight decay than wider models. It seems thus natural to choose a gaussian prior with small variance i.e. more restrictive for narrower networks than for wider ones.

The fact that the author(s) have not optimized their training protocol might partially explain the (possible) discrepancy with other works: see for example  https://arxiv.org/abs/1901.01608 (which should be cited and discussed), ref 49, where it is explicitly shown that width improves accuracy *in practice*.

Indeed, the quality of an architecture is not only in its theoretical expressivity, but, even more, in its “trainability”. Even a large  width perceptron is, as has been well known since a long time, extremely “expressive”, but it cannot be trained.  The importance of trainability is in my opinion overlooked in an otherwise interesting and relevant paper.

In short, I think that either the detrimental effect of width is demonstrated using state-of-the-art training protocols, or the strong claims on the importance of Theorem 1 for standard DNNs should be weakened.

 The code used for the experiments should be publicly released for the reproducibility of the results


**Time Spent Reviewing:**

6

---

> ### Author Response · Authors · 2021-08-09
> **Reviewer zKHo**
>
> Thank you for your constructive remarks. We will address your concerns about the experiments in Figure 4, and we are pleased that you find the rest of the paper “interesting and relevant.”
>
> > “The test accuracy reported in Figure 4, rightmost panel, is significantly lower than the accuracy obtained with non-Bayesian approaches on CIFAR10. This is for sure partially due to the lack of data augmentation (which is acceptable in a numerical experiment) but, in my opinion, also to the training protocol.”
>
> We would argue that this discrepancy can almost entirely be explained by the lack of data augmentation. [Huang et al (2017, Table 2)](https://arxiv.org/pdf/1608.06993.pdf) report 86.4% accuracy on *un-augmented* CIFAR10 using a ResNet-110 model, which is very similar to the accuracy that we report in Figure 4 using wide ResNet-20 models. Moreover, our training protocol (see Appendix H) exactly matches the (previously SOTA) protocol outlined by He et al. [42].
>
> > “The fact that the author(s) have not optimized their training protocol might partially explain the (possible) discrepancy with other works: see for example https://arxiv.org/abs/1901.01608 (which should be cited and discussed), ref 49, where it is explicitly shown that width improves accuracy in practice.”
>
> Thank you for making us aware of the Geiger et al. (2019) paper, which we will of course cite. We note however that their training procedure uses full batch gradient descent with learning rates on the order of 1e-4, whereas we use (minibatched) Adam with standard learning rates. Therefore, it is reasonable that we arrive at different conclusions. Moreover, the results in [49] in fact suggest that narrower networks can outperform wider ones (see the full table in [49, Appx. D], where the best performing NN often have width <2000).
>
> > “If I understood correctly, the same weight decay is applied for every model. What would happen if one would tune this decay optimally?... It seems thus natural to choose a gaussian prior with small variance i.e. more restrictive for narrower networks than for wider ones.”
>
> The  reasoning behind our weight decay scheme is to see how width affects model performance while keeping the corresponding prior on the NN parameters constant. Choosing a width-specific width decay would in some sense be performing the opposite analysis - i.e. determining what prior is best for a given width, rather than the other way around. Nevertheless, we do agree that it makes sense to explore these results with different levels of weight decay. For N=5000 MNIST, a width-256 network performs best with 1e-5 weight decay (95.5%), width-1024 performs best with 1e-6 (95.7%), and width-2048 performs best with 1e-7 (95.9%). We will include additional results in the final version.
>
> > “The importance of trainability is in my opinion overlooked… the limitations are not sufficiently discussed, in particular concerning the connection of the results to the theory of DNNs.”
>
> We agree that there are important factors of (non-Bayesian) neural networks that are not addressed by our theory, some of which are addressed in Section 7 (lines 316-320). We note that these practical factors are in some sense orthogonal to the questions asked by this paper, which primarily aim to study the effects of width on model priors/posteriors, decoupled from practical concerns about computing these posteriors/training these models. Nevertheless, we concur that the notion of “trainability” is indeed of practical importance, and something we hope to study in future work.

---

> > ### Comment · Reviewer_zKHo · 2021-08-28
> > **Thanks for your response, and some more comments**
> >
> > I see that also reviewer h2Yp had concerns on the experimental results, which were obtained by specific hyperparameters. I also see that some dedicated experiment has been conducted to study the effect of weight decay (as I asked) and that the results will be presented in the final version. I  am puzzled by  your reply to my one of my comments: "We note however that their training procedure uses full batch gradient descent with learning rates on the order of 1e-4, whereas we use (minibatched) Adam with standard learning rates. Therefore, it is reasonable that we arrive at different conclusions. " This is exactly what I try to say: the dependence of the accuracy on the width is determined by the training protocol. I believe that the theoretical results presented in the manuscript are  valuable, and I will "vote" for publication, but I insist  that either the detrimental effect of width is demonstrated by extensive and accurate experiments (which I see as beyond the scope of the paper), or the strong claims on the importance of Theorem 1 for standard DNNs should be weakened.

---

> > > ### Author Response · Authors · 2021-08-28
> > > **How we will incorporate your feedback**
> > >
> > > Thank you for taking the time to follow up and further engage with our paper. We agree with your suggestions, and below we briefly outline how they will be incorporated in the final version:
> > >
> > > > I believe that the theoretical results presented in the manuscript are valuable, and I will "vote" for publication, but I insist that either the detrimental effect of width is demonstrated by extensive and accurate experiments (which I see as beyond the scope of the paper), or the strong claims on the importance of Theorem 1 for standard DNNs should be weakened.
> > >
> > > We will tone down the language in the paper about the connection of our results to conventional neural networks (e.g. lines 18-19 and 310-311). We will increase our discussion in Section 7 (paragraph 2) of other works that draw different conclusions about the effect of width, noting that hyperparameters like learning rates, initializations, and weight priors have an effect not captured by our theory.
> > >
> > > We stand by the validity of our results in Section 6.2 (and the follow-up hyperparameter sweep); nevertheless, we hope that this change in tone and discussion of limitations sufficiently incorporates your feedback.

---

### Official Review · Reviewer_dRMh · 2021-07-16

**Rating:** 7
**Confidence:** 3

**Summary:**

     This paper provides a narrative of composite Gaussian process models. In specific the paper provides insights into the interplay between depth and width in the context of capacity and representative power. The paper further draws conclusions on the implications that this has for composite finite basis function models such as neural networks.


**Limitations And Societal Impact:**

Yes

**Main Review:**

This is a very nice paper that I thoroughly enjoyed reading. The paper really tries to tell a story through related work and provides easy to follow mathematical derivations of the main concepts with plenty of intuition.

     My main question regarding the generalisation of the results from Deep GPs to neural networks. While it is true that Deep GPs subsumes neural networks and as is shown in [1] there exists an equivalence class in parametrisation between a sparse Deep GP and a neural network. However, the main difference comes in the difference between the marginal likelihood and the objective function used to train the neural network. It would be great if the authors could comment on this and what the implications of this is in interpreting the results in the paper for neural networks.

     While this paper wants to study the model in isolation from the inference mechanism I think the results implies something quite interesting with respect to sparse DGPs, the approximate model is actually more useful compared to the model it aims to approximate. The same type of behaviour can be seen with a VAE. I think this would be an interesting idea for the authors to discuss and what this actually means for the conclusions we should draw from inference and how we should design approximate inference schemes for Deep GPs.

     In summary I think this is a lovely written paper providing a nice narrative to many concepts that people have had an intuitive understanding of with respect to Deep GPs. Making these mathematically concrete and providing a narrative grounded in related work is important to move these methods forward with understanding beyond empirical results. The authors have done a good job of separating the material in the main paper from the detailed explanations in the appendix. For me it was crucial to have the verbose derivations in Appendix F,G to solidify my understanding from the paper. I would be happy to increase my score, however I feel (as is reflected by my confidence score) that I am less well versed in the related work that this paper builds upon so there might be things that I have missed.

     Even though this paper has *a lot* of references already, one paper that the authors might like is [2]. There are quite a few connections between these two works that you might find interesting for future work.

     1) Dutordoir, V., Hensman, J., Wilk, M. v. d., Ek, C. H., Ghahramani, Z., & Durrande, N., Deep neural networks as point estimates for deep gaussian processes, CoRR, (),  (2021).
     2) Halverson, J., Maiti, A., & Stoner, K., Neural networks and quantum field theory, Machine Learning: Science and Technology, 2(3), 035002 (2021).  http://dx.doi.org/10.1088/2632-2153/abeca3

**Time Spent Reviewing:**

5

---

> ### Author Response · Authors · 2021-08-09
> **Response to Reviewer dRMh**
>
> Thank you for your comments. We appreciate the reference to Halverson et al. (2021) and will reference it in the final version.
>
> > “[Please remark on] differences between the marginal likelihood and the objective function used to train the neural network.”
>
> This is an important point, and we will expand our discussion of it in Section 6.2. The standard objective function for neural networks is usually some probabilistic loss function (MSE or cross entropy) coupled with weight decay regularization, which of course can be interpreted as a MAP objective function for a neural network with a Gaussian prior. This is different from a Bayesian treatment of NN parameters (as explored in Section 6.1), and complexities from the non-convex optimization problem may also affect the results. Nevertheless, we find that our analysis of the Bayesian setting seems to be predictive of performance in the non-Bayesian (MAP) setting, and may offer insight into the inductive biases of NN.
>
> > “... what [does] this actually mean for the conclusions we should draw from inference and how we should design approximate inference schemes for Deep GPs?”
>
> Thank you for this interesting suggestion. A sparse Deep GP is still a Deep GP, albeit one with low-rank covariance function, and so most of our analysis on width should hold for sparse Deep GP priors. However, it is not immediately clear how width affects models with approximate inference schemes like VI, and this is something we hope to investigate in future work.

---

### Official Review · Reviewer_h2Yp · 2021-07-19

**Rating:** 5
**Confidence:** 4

**Summary:**

The authors show that in the infinite width limit, deep GPs becomes a single-layer GP, generalizing the result that deep neural networks at initialization become a GP in the infinite width limit. The authors show that increasing depth of the GP makes the distribution of the final layer of deep GPs deviate from being Gaussian at finite width. The authors claim that this simplification implies that large widths are detrimental to neural network performance beyond some point.

**Limitations And Societal Impact:**

I do not have useful comments in this aspect.

**Main Review:**

### Originality
The authors study the effect of the width and depth of deep GPs (a generalization of deep networks at initialization) and discuss the implications of the depth and width on the distribution of the output layer conditioned on the input features. This is an interesting generalization of the results that deep neural networks in the infinite limit become GPs at initialization.

### Quality
The quality of the theoretical analysis of the work is good. The experiments conducted on neural networks, and the authors' assertion that larger widths hurts neural network performance (because they make networks into GPs) has the following flaws:

1-0. In order to conduct a fair comparison of the performance of two neural networks with different width, one needs to conduct hyper-parameter search of basic parameters. At a minimum, it is standard to at least conduct a hyper-parameter search of the learning rate, which the authors do not do. This brings into question the validity of figure 4, because, for example, the performance of a width 5000 network might be worse than the performance of a width 500 network, because the fixed learning rate the authors used is more optimal for the width 500 network.

1-1.  For example, it is very possible that the drop-off of performance with respect to the width of the network for MNIST shown in figure 4 can be related to stability issues, as the maximum stable learning rate for wider networks tend to be lower. We can't know for sure, since the authors have not conducted a learning rate sweep for each width.

2-0. Besides the fact that the experiments conducted to support the author's assertion that "very large" widths hurt neural network performance has not been carefully conducted, even if one accepts the assertion that there exists an optimal width for performance, the reason the authors provide for it is not consistent with the existing literature.

2-1. If the authors' diagnosis for width hurting performance, i.e., that the neural network collapses to a GP in that limit, is correct, it must be the case that the performance of a neural-network GP (NNGP) cannot exceed that of a neural network for the MNIST tasks the authors explored.

2-2. This can be refuted by the experiments presented in work the authors cite, namely Lee et al. [49]. In table 1 and table 2 of their paper, the authors present the result of numerous experiments they conduct where they demonstrate examples where the NNGP performance exceeds that of a trained neural network. For example, the authors report an NNGP (obtained as the infinite width limit of a 3-layer network) that achieves 96.93% test accuracy on 5k MNIST, better than what the authors report in figure 4.

### Clarity
The theoretical analysis itself has been presented in a clear and consistent way. The connection between the theoretical results and the implication it has to the performance of neural networks has not been clearly presented, if presented at all. The authors try to imply that the fact that the infinite limit reduces the neural network at initialization to a GP (a well-known fact) is the reason that very wide networks should eventually under-perform networks with "optimal-width," but there is hardly an argument being made. In fact, Lee et al. [49] has systematically explored this problem with extensive experiments and have showed that the GP limit often out-performs trained neural networks, disproving the authors' claim.

### Significance
The paper contains some interesting analysis on deep GPs, but their claims connecting their theoretical results to the performance of actual neural networks are not consistent with experimental results in the literature.

---

### Summary of Author Correspondence

After discussion with the authors, the authors have agreed to make the following changes to their paper:
1. They have run additional experiments to show how widening the width of trained-NNs affect the NN performance, and agreed to provide the training parameters used.
2. They have agreed to clarify that their theoretical and experimental results on deep GPs cannot be directly be applied to trained-NNs, but is suggestive in nature.

My initial rating of the paper for a clear rejection (3) was based on a (perhaps misunderstood) assessment on my part that the paper is confusing the reader by implying that the Gaussianity of wide DGP marginals is what should be attributed to the performance loss of trained-NNs at very large widths, the evidence of which I found to be unconvincing. Given that the authors have agreed to clear up such confusion, I believe my initial rating to be too harsh now. At the same time, it has been clarified that the most interesting part of the paper to me, the consequence of the authors' work on trained-NNs, is indeed of a speculative nature. I thus have adjusted my rating to a marginal reject (5).

**Time Spent Reviewing:**

4

---

> ### Author Response · Authors · 2021-08-09
> **Response to Reviewer h2Yp**
>
> > “In order to conduct a fair comparison of the performance of two neural networks with different width, one needs to conduct hyper-parameter search of basic parameters... which the authors do not do.”
>
> We performed a more extensive hyperparameter search and found no changes to our main conclusions. We will update all results in the final version, but for brevity we will describe results for the MNIST N=5000 experiment. We test 3-layer models (2 hidden layers) with various weight decay values (1e-5 to 1e-7) and we perform a random search over learning rate/minibatch size. Our results still hold: the best performing model is a width-2048 model with 1e-7 weight decay, achieving 95.97% error - narrower than the largest model we test. (Note that this number differs from the 96.93% number you report, which comes from a network with 3 hidden layers, tanh activations, and a regression loss.)
>
> > “If the authors' diagnosis for width hurting performance... is correct, it must be the case that the performance of a neural-network GP (NNGP) cannot exceed that of a neural network for the MNIST tasks the authors explored.”
>
> As we state in the general remarks, comparing NNGP directly to trained NN conflates the effects of width with the choice of inference method, and therefore NNGP outperforming trained (non-Bayesian) NN does not invalidate our theoretical results. Indeed, when we compare NNGP to *Bayesian* neural networks (Figures 3 and 9, orange lines) - thus controlling for the inference method - we find that finite width almost always outperforms infinite width. The major takeaway from our trained NN experiments is that there is a limit to the benefit of width. We note that *this finding is in fact supported by [49, Table 2],* which shows that - on more than half of the tasks - the best performing NN has width <= 2000, narrower than the widest NN tested.

---

> > ### Comment · Reviewer_h2Yp · 2021-08-26
> > **Thank you for your response**
> >
> > Thank you for your response. I have added my response below.
> >
> > ### Comment 1
> > Thanks for conducting an extended hyper-parameter search for the "maximal performance" experiment. I think this is a key result in the paper, and sparse information has been given on achieving the numbers shown in the plot. I still think the authors should provide better details on the experiments conducted rather than claim that a "random search has been conducted." The numbers shown depend as much on the training hyperparameters used as on the width itself. In addition, when varying the width, the regime of "best hyperparameters" for the network shifts, and the random search window needs to be shifted.
> >
> > To explain a bit more, it is understood that for larger widths, the range of stable learning rates scale like $1/\sqrt{\text{width}}$. Meanwhile, the performance of the network is determined by the combination of (learning rate)/(batch size) thus the regime of batch sizes explored should scale down as the learning rate is scaled down with width. It is unclear if the authors have followed this practice for their hyperparameter search.
> >
> > In fact, there have been suggestions in the literature that large network performance plateaus/degradation can come from this phenomenon, i.e., constraints of learning rate stability bounds and the fact that the batch size is bounded below by 1.
> >
> > ### Comment 2
> > I understand that comparing NNGP directly to trained NN is not completely fair. I would say, however, that this data point, i.e., that NNGPs out-performed trained-NNs should give the authors pause when they try to attribute the wide-trained-NN performance degradation they observe (which I think needs further details for one to be convinced about) to the DGPs becoming more Gaussian in the wide limit.
> >
> > Also, if one is to argue that comparing the trained wide-NN performance to NNGP performance is unfair due to the choice of inference method, the lessons learned from observing the DGP marginals should not be expected to be directly applied to trained wide-NNs, unless with further evidence. A convincing result would have been if the trained-NN plot peaked at a higher performance than the NNGP and then asymptoted to the NNGP performance at larger widths. This is the reason for the suggestion of conducting better hyperparameter search for the trained-NNs, for the hope that this kind of phenomenon could be observed to support the authors' thesis.
> >
> > ### Final Comment
> > I would like to re-iterate that while the authors' analysis of the wide-limit of DGPs are well-founded and the experiments have been clearly presented, their presentation of the trained-NN experiments and explanation of the wide-width limit behavior of these networks is unsatisfactory. This is unfortunate, because the trained-NN results are the most consequential portion of the paper for the general practitioner.

---

> > > ### Author Response · Authors · 2021-08-27
> > > **Follow up**
> > >
> > > ## Comment 1
> > >
> > > > I still think the authors should provide better details on the experiments conducted rather than claim that a "random search has been conducted." The numbers shown depend as much on the training hyperparameters used as on the width itself.
> > >
> > > We use random search as our hyperparameter optimization strategy. (We note that this is the same strategy used in Lee et al. 2018.) In particular, we selected 20 learning rate/batch size tuples from $[0.001, 0.2] \times \{16, 32, 64, 128, 256 \}$, and trained networks with these hyperparameter settings. We then select the network hyperparameters that achieve the best validation set accuracy. We perform this optimization for each network width/weight-decay, so we are choosing width-specific hyperparameters.
> > >
> > > > It is understood that for larger widths, the range of stable learning rates scale like $1 / \sqrt{\text{width}}$.
> > >
> > > Would you be able to provide a citation for this? There are a number of successful empirical works where the learning rate is not scaled with width. For example, [Zagoruyko et al. 2016](https://arxiv.org/abs/1409.1556) train networks of various width using the same learning rate/batch sizes parameters, and their results suggest no issues with stability. Moreover, the experiments in Lee et al. (2018) use the same learning rate/batch size search space for networks of all widths.
> > >
> > > > Meanwhile, the performance of the network is determined by the combination of (learning rate)/(batch size) thus the regime of batch sizes explored should scale down as the learning rate is scaled down with width. It is unclear if the authors have followed this practice for their hyperparameter search.
> > >
> > > The best performing learning rate for all models was > 0.005. This suggests that our learning rate search space was adequate.
> > >
> > > > In fact, there have been suggestions in the literature that large network performance plateaus/degradation can come from this phenomenon, i.e., constraints of learning rate stability bounds and the fact that the batch size is bounded below by 1.
> > >
> > > We would appreciate a reference to the literature that you are referring to. We are not sure whether these findings would apply to our hyperparameter-sweep experiment however, as the largest network we explore is width 8192, which is a regime where standard NN hyperparameters have been shown to successfully train networks without stability issues (see e.g. [Simonyan and Zisserman, 2015](https://arxiv.org/abs/1409.1556)).
> > >
> > > ## Comment 2
> > >
> > > > Also, if one is to argue that comparing the trained wide-NN performance to NNGP performance is unfair due to the choice of inference method, the lessons learned from observing the DGP marginals should not be expected to be directly applied to trained wide-NNs, unless with further evidence.
> > >
> > > We agree. We are not claiming that our theory on Bayesian models should *directly* apply to conventional NN, but rather that “these [DGP/Bayesian NN] results make strong predictions about the same phenomenon in conventional neural networks.” It is reasonable to assume that there should be some connection between Bayesian networks and conventional ones, since the training objective of conventional networks is the MAP objective of Bayesian neural networks. This is what we confirm in Figure 4: empirically there is a similarity between Bayesian models (what we study theoretically) and conventional models with regards to width.
> > >
> > > We would also reiterate that we do not compare Bayesian NN/DGP against conventional NN; but rather use the Bayesian NN/DGP results to motivate an investigation of conventional NN.
> > >
> > > > A convincing result would have been if the trained-NN plot peaked at a higher performance than the NNGP and then asymptoted to the NNGP performance at larger widths. This is the reason for the suggestion of conducting better hyperparameter search for the trained-NNs, for the hope that this kind of phenomenon could be observed to support the authors' thesis.
> > >
> > > We believe that you may be misunderstanding our thesis. The point of our paper is not to compare finite-width trained neural networks against NNGP models (this is the point of Lee et al. (2020), who do so through extensive experiments). The purpose of our paper is to 1) understand how width affects Bayesian neural networks and DGP and 2) see if this theory/results are indicative of similar phenomenon in non-Bayesian neural networks. We believe that the experiments that we have performed are sufficient with regards to these research questions.
> > >
> > > ## Final note
> > >
> > > > I would like to re-iterate that while the authors' analysis of the wide-limit of DGPs are well-founded and the experiments have been clearly presented, their presentation of the trained-NN experiments and explanation of the wide-width limit behavior of these networks is unsatisfactory.
> > >
> > > We have supported our original findings with a more extensive hyperparameter search, and we reiterate that these findings are supported by other empirical works (see “general response to the reviewers”). We hope that you can specifically clarify what improvements you think are needed with regards to this result.

---

> > > > ### Comment · Reviewer_h2Yp · 2021-08-27
> > > > **Thank you for your follow up**
> > > >
> > > > Thank you for your follow up. See comments below.
> > > >
> > > > ### Comment 1
> > > > On hyperparameter tuning for training neural networks with varying widths, there are a few extensive studies that provide ample experimental details. You can see for example, [R1]. You can see some related analysis in [R2].
> > > >
> > > > Again, it would have been great if the presentation of the results for the NN training given in the original work and your response would have been more specific (e.g., for the random search you've conducted, specify the range of parameters or the number of experiments for the random search) to help the common practitioner to interpret the result. It is my opinion that the amount of information provided for the experiments make it hard to judge which factor should be attributed to the decreasing performance for larger widths.
> > > >
> > > > I agree that what you have observed can be a genuine phenomenon, but again, my main issue is the lack of basic experimental details provided (both in the initial paper and your follow up response) to give the reader context.
> > > >
> > > > ### Comment 2
> > > > It would be great if you would be able to make clear that the relation between your result with "conventional neural networks" to be of a speculative nature rather than a logical extension of the rest of the paper, which I think can be a cause of confusion for the reader. To myself, such a relation would be the most consequential part of the paper, since it would provide a theoretical basis for explaining an interesting phenomenon.
> > > >
> > > > For example, in the abstract you state that your results "make strong predictions about the same phenomenon in conventional neural networks." If you are ready to dismiss the NNGP performance having anything to do with trained-NN inference accuracy, I do not think you can claim that the marginal of a DGP approaching Gaussianity at larger width "makes strong predictions about" the trained NN having an optimal width for performance, especially when a Gaussian network can be used to achieve better inference accuracy than the trained network. From our discussion, these seem to be two interesting phenomena that may be related, rather than one strongly implying the other.
> > > >
> > > > ### Final Comment
> > > > Thanks again for taking the time for following up. I feel the paper would be greatly improved and would be able to give a clearer picture to the reader of the consequence of the work that has been done if the above comments were taken into account.
> > > >
> > > >
> > > > [R1] Park et al., "The Effect of Network Width on Stochastic Gradient Descent and Generalization: an Empirical Study."
> > > >
> > > > [R2] Lewkowycz et al., "The large learning rate phase of deep learning: the catapult mechanism."

---

> > > > > ### Author Response · Authors · 2021-08-27
> > > > > **Thank you for your detailed responses.**
> > > > >
> > > > > Thank you for your detailed responses. We would like to clarify a few points, and reiterate our outline for how we will address your concerns.
> > > > >
> > > > > > Again, it would have been great if the presentation of the results for the NN training given in the original work and your response would have been more specific (e.g., for the random search you've conducted, specify the range of parameters or the number of experiments for the random search) to help the common practitioner to interpret the result. I agree that what you have observed can be a genuine phenomenon, but again, my main issue is the lack of basic experimental details provided (both in the initial paper and your follow up response) to give the reader context.
> > > > >
> > > > > What experimental details are missing? We have provided the hyperparameter search details in our discussion, and we will of course add them to the final version. All other experimental details can be found in Appendix H (or the accompanying code, which has also been provided and will be publicly released upon publication). This should be enough for experimental analysis and reproducibility.
> > > > >
> > > > > > It would be great if you would be able to make clear that the relation between your result with "conventional neural networks" to be of a speculative nature rather than a logical extension of the rest of the paper, which I think can be a cause of confusion for the reader… for example, in the abstract you state that your results "make strong predictions about the same phenomenon in conventional neural networks."
> > > > >
> > > > > We will adjust the language of the abstract and Section 6.2 accordingly. We will make it more clear that our theoretical results do not directly apply to conventional neural networks, and that we are empirically measuring to what extent the phenomenon we identify in Bayesian models also occurs with conventional neural networks. We will also include additional discussion about how our work relates to other empirical results (building upon what we have in Section 1, paragraph 2).
> > > > >
> > > > > > Thanks again for taking the time for following up. I feel the paper would be greatly improved and would be able to give a clearer picture to the reader of the consequence of the work that has been done if the above comments were taken into account.
> > > > >
> > > > > We believe that we have sufficiently outlined a way to improve the paper based on your comments: namely, 1) including the details about the hyperparameter search and 2) toning down the language regarding the “strong predictions about the same phenomenon in conventional neural networks.” These are the primary issues that you bring up in your review/follow up, and we hope that these concrete steps will address your concerns. If so, we ask that you consider raising your score.

---

> > > > > > ### Comment · Reviewer_h2Yp · 2021-08-27
> > > > > > **Re: Thank you for your detailed responses.**
> > > > > >
> > > > > > Thank you for your prompt reply.
> > > > > >
> > > > > > ### Comment 1
> > > > > > Sorry for missing the hyperparameters you provided in your second response. While I believe the range probed is too narrow (and possibly biased against wider networks), it would be helpful to add that information to the paper so the reader can make their own judgement.
> > > > > >
> > > > > > ### Comment 2
> > > > > > Thank you for agreeing to adjusting the language in the paper. I believe by doing so, you would be able to avoid confusion among readers like myself.
> > > > > >
> > > > > > Since you have agreed to make these changes, I will adjust my scores accordingly.

---

### Author Response · Authors · 2021-08-09
**General response to the reviewers**

We thank the reviewers for their constructive and insightful feedback. Overall, we are pleased that reviewers found our work “important” (dRMh), “well written” (zKHo), and “strong and noteworthy” (PVcF). The notable exception is the review of h2YP, which questions the validity of our results. We will engage in detailed technical discussions with each reviewer individually, but we would like to begin by summarizing high-level concerns and addressing the claims of h2YP.

Several reviewers note that our trained NN results in Figure 4 draw different conclusions than previously published work (namely [49], [51], and [Geiger et al. 2019](https://arxiv.org/abs/1901.01608)). While we acknowledge these differences and indeed believe these differences emphasize the value of our theory, **we point out that there are several empirical works that also support our conclusion.** For example, although experiments by Geiger et al. show that width is beneficial in the full-batch low-learning rate regime, [Allen-Zhu et al. [3]](https://arxiv.org/pdf/1905.10337.pdf) empirically demonstrate that narrow ResNet outperform wider ones. Similarly, [Aurora et al. [7]](https://arxiv.org/pdf/1904.11955.pdf) show that finite CNN with global pooling outperform their infinite-width counterparts -- a finding that is validated by [51]. We argue that these results do not conflict with one another, but rather build upon each other by exploring different facets of this complex phenomenon that is quite new in the literature. Please see the individual comments for more discussion.

Finally, we respectfully reject the premise that “Lee et al. [49]... have showed that the GP limit often out-performs trained neural networks, disproving the authors' claim.” (h2YP).  First, a single paper’s empirical results, especially in this relatively new literature, simply do not close the book on this question, nor do they set a gold standard (despite that paper being a nice piece of work).  Second, as previously noted, our results on the effect of width are supported by other more recent works [e.g. 3, 7].  Third and perhaps more importantly, **one cannot draw conclusions about the effect of width from [49]’s comparison of (Bayesian) NNGP and (non-Bayesian) trained NN, as these model classes differ in the choice of inference method (Bayesian inference versus optimization).** We discuss this point more in the individual remarks.

---

### Decision · Program_Chairs · 2021-09-27

**Decision:**

Accept (Poster)

**Comment:**

This paper leverages an analysis of deep Gaussian processes to argue that an excess increase in the width of a neural network can degrade performance. The question of whether or not neural networks benefit from increased width is an important and unresolved question in the literature, and the reviewers and I agree that this paper provides an additional and important perspective on the topic that will be of interest to the NeurIPS community. While some reviewers found the experimental results unconvincing and the conclusions somewhat speculative, others found the framework to be illuminating and to provide new perspectives on this important problem. Overall, I do not expect this or any paper to fully and unambiguously resolve all questions relating to the benefits/drawbacks of large width networks, and I believe this paper provides novel and useful insights that shed light on this important problem. Therefore, I recommend acceptance.